# Mlf mediates proteotoxic response via formation of cellular foci for protein folding and degradation in *Giardia*

**Martina Vinopalová**[1], **Lenka Arbonová**[1], **Zoltán Füssy**[2], **Vít Dohnálek**[1], **Abdul Samad**[1], **Tomáš Bílý**[3], **Marie Vancová**[3], **Pavel Doležal**[1]*

**1** Department of Parasitology, Faculty of Science, Charles University, BIOCEV, Prague, Czech Republic, **2** Scripps Institution of Oceanography, University of California San Diego, La Jolla, California, United States of America, **3** Institute of Parasitology, Biology Centre of the Academy of Sciences of the Czech Republic, Faculty of Science, University of South Bohemia, České Budějovice, Czech Republic

* pavel.dolezal@natur.cuni.cz

**Data Availability Statement:** All data are in the manuscript and supporting information files. The submission contains all raw data required to replicate the results of your study.

## Abstract

Myeloid leukemia factor 1 (Mlf1) was identified as a proto-oncoprotein that affects hematopoietic differentiation in humans. However, its cellular function remains elusive, spanning roles from cell cycle regulation to modulation of protein aggregate formation and participation in ciliogenesis. Given that structurally conserved homologs of Mlf1 can be found across the eukaryotic tree of life, we decided to characterize its cellular role underlying this phenotypic pleiotropy. Using a model of the unicellular eukaryote *Giardia intestinalis*, we demonstrate that its Mlf1 homolog (GiMlf) mainly localizes to two types of cytosolic foci: microtubular structures, where it interacts with Hsp40, and ubiquitin-rich, membraneless compartments, found adjacent to mitochondrion-related organelles known as mitosomes, containing the 26S proteasome regulatory subunit 4. Upon cellular stress, GiMlf either relocates to the affected compartment or disperses across the cytoplasm, subsequently accumulating into enlarged foci during the recovery phase. *In vitro* assays suggest that GiMlf can be recruited to membranes through its affinity for signaling phospholipids. Importantly, cytosolic foci diminish in the *gimlf* knockout strain, which exhibits extensive proteomic changes indicative of compromised proteostasis. Consistent with data from other cellular systems, we propose that Mlf acts in the response to proteotoxic stress by mediating the formation of function-specific foci for protein folding and degradation.

## Author summary

*Giardia intestinalis*, a widespread intestinal parasite, has emerged as a valuable model for studying eukaryotic cell biology and host-pathogen interactions. Our study focuses on the Myeloid Leukemia Factor (Mlf) homolog in *Giardia* (GiMlf), an evolutionarily conserved protein with diverse cellular functions. We demonstrate that GiMlf plays a crucial role in proteostasis by forming two types of cytosolic foci: one associated with microtubular structures and Hsp40, and another comprising ubiquitin-rich, membraneless

**Funding:** This work was supported by project START (START/SCI/012, MŠMT), Project name: Grant schemes at Charles University, Reg. number: CZ.02.2.69/0.0/0.0/19_073/0016935) to MVi, LA and VD and by a grant from Czech Science Foundation 20-25417S to PD. The funders had no role in study design, data collection and analysis, decision to publish, or preparation of the manuscript.

**Competing interests:** The authors have declared that no competing interests exist.

compartments near mitosomes, containing the 26S proteasome regulatory subunit. Upon cellular stress, GiMlf dynamically relocates, suggesting its involvement in the stress response. Notably, GiMlf knockout leads to extensive proteomic changes and altered encystation rates, indicating its importance in *Giardia's* life cycle and stress adaptation. Our findings provide insights into how *Giardia*, and potentially other eukaryotes, maintain proteostasis under various environmental conditions. This research enhances our understanding of fundamental parasite biology and stress response mechanisms, which are critical for pathogen survival in diverse host environments.

## Introduction

Mlf1 is a soluble protein of nuclear and cytosolic localization, which was originally identified in chromosomal translocations that lead to acute myeloid leukemia [1]. Later, the protein has been implicated in a variety of seemingly unrelated functions such as the cell cycle exit, differentiation, transcription, apoptosis, and cell proliferation [2]. In human cells, Mlf1 was specifically reported to influence the stability of p27Kip1 [3] and p53 by inhibiting their degradation by the proteasome [4], it participates in the regulation of ciliogenesis [5], and patients with mutated Hsp40 (DNAJB6) were found to accumulate Mlf1 in clusters with Hsp40 [6] In DYT1 dystonia, nonfunctional Torsin ATPase causes defects in the nuclear pore assembly, and the Mlf1 paralog Mlf2 relocalizes along with Hsp40/Hsp70 proteins to the nuclear envelope blebs [7,8]. Other independent studies have attributed various functions to Mlf proteins, yet a unifying role remains rather elusive.

*Giardia intestinalis* (syn. *duodenalis*, *lamblia)* is a unicellular eukaryote (protist) that colonizes the walls of the small intestine of various vertebrates and causes giardiasis, the most common intestinal parasitic disease in humans worldwide [9]. Because of its unique cell structure and the availability of genetic approaches, *Giardia* has become a valuable cell biology model. It has a highly developed microtubule (MT) cytoskeleton with eight flagella and a prominent adhesive disc, by which it attaches to the intestinal epithelia. Furthermore, some typical eukaryotic organelles, such as the Golgi apparatus and peroxisomes, have not been convincingly demonstrated in *Giardia* [9–11]. The so-called mitosomes, enclosed by a double membrane, represent highly reduced mitochondria whose sole function is the assembly of Fe-S clusters [12]. In the intestine, *Giardia* undergoes differentiation into infectious cysts through a process called encystation, which can be induced *in vitro* [13]. During encystation, the cyst wall proteins (CWPs), which form the cyst wall, are transported to the cell surface via encystation-specific vesicles (ESVs) [14–16] that associate with subunits of the 26S proteasome [17].

Several studies have found a *Giardia* homolog of Mlf1 (GiMlf) at different cellular locations such as mitosomes [18,19], cytoskeletal elements [20], or within so-called MLF vesicles (MLFVs) [21,22]. Additionally, in large-scale analyses, GiMlf was shown to respond to elevated temperature, the anti-giardial drug metronidazole [23,24], and other small inhibitors [22].

These data, along with information from other eukaryotes, suggest a highly diverse functional repertoire for Mlf proteins. We therefore set out to elucidate the unifying function of Mlf that could explain its phenotypic diversity. In this study, we confirmed that Mlf represents an ancient eukaryotic protein predating the last eukaryotic common ancestor (LECA) and is ubiquitously found in metazoans, plants, and protists. Using the *Giardia* model, we delineated the cellular roles of Mlf. Under physiological conditions, GiMlf localizes to two distinct types of cytosolic foci. Specifically, it interacts with Hsp40 at microtubule cytoskeleton structures, and with the regulatory subunit 4 of the 26S proteasome (P26s4) near mitosomes. Our data

suggest that these foci represent membraneless ubiquitin-rich compartments whose stability depends on the presence of GiMlf. Upon exposure to thermal stress, GiMlf expression is elevated, leading to its dispersal from P26s4-positive foci throughout the cell and subsequent accumulation within enlarged foci. During the overexpression of organelle-specific membrane proteins, GiMlf relocalizes to the affected compartment, and the *in vitro* data indicate that this could be mediated by the recognition of signaling phospholipids on the cytosolic face of the membranes. Proteomic analysis showed that the absence of GiMlf causes an extensive proteostatic imbalance. In alignment with recent reports, this study proposes that Mlf proteins act in the response to proteotoxic stress and are also involved in the formation of specialized foci dedicated to protein folding and degradation, thus underscoring their critical role in maintaining cellular homeostasis.

## Results

### Mlf is an evolutionarily conserved protein that was present in LECA

Given the constrained taxon sampling, we performed a hidden Markov model (HMM)-based homology search to identify the presence of Mlf orthologs across the eukaryotic tree of life. In addition to the already known Mlf proteins from Metazoa and Metamonada, clear orthologs could be identified in other metamonad species such as trichomonads, *Spironucleus salmonicida* and other eukaryotic supergroups, including Archeaplastida, Amoebozoa, Cryptista, Haptista, SAR, and Excavata (Fig 1A). The phylogenetic analysis of the sequences suggested that the Mlf protein family existed in LECA but was secondarily lost in some lineages, such as Fungi and Apicomplexa (Figs 1A and S1). Conversely, in some eukaryotes such as Metazoa,

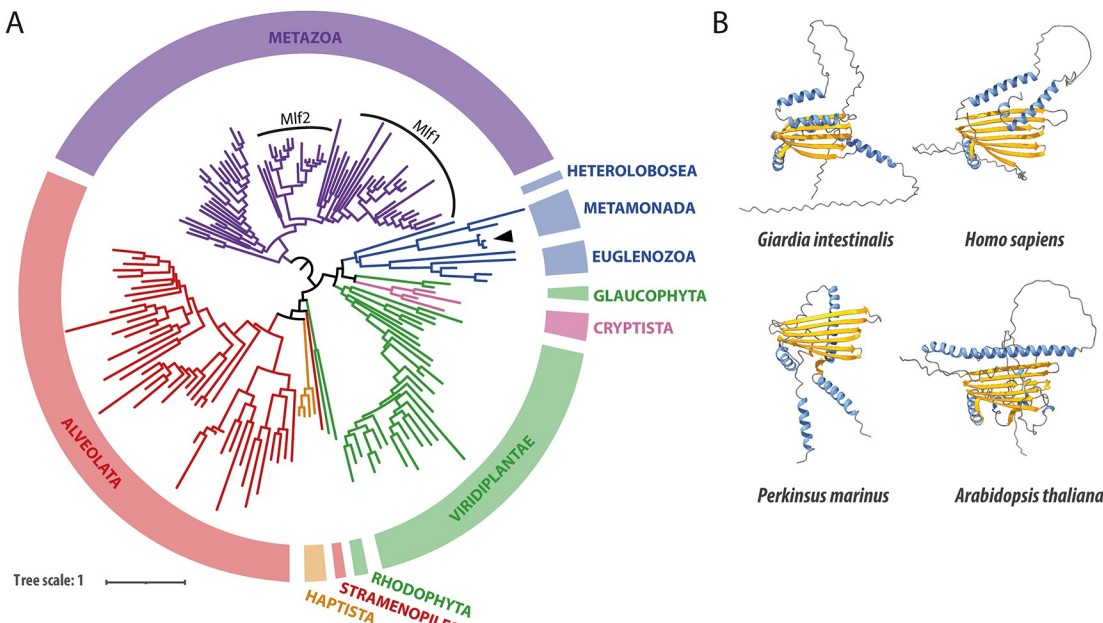

**Fig 1. GiMlf is a structurally and evolutionarily conserved protein whose homologs are present among the majority of eukaryotic supergroups. (A)** Phylogenetic tree of GiMlf homologs generated by IQ-TREE based on a MAFFT alignment after trimming with TrimAI (discarded positions with >30% missing data). Major eukaryotic supergroups are highlighted by colors (Obazoa—purple, Excavata–blue, Archeaplastida–green, Cryptista—pink, Haptista–orange, SAR–red). The arrowhead labels the position of GiMlf within the tree. Two paralogs Mlf1 and Mlf2 are recognized within Metazoa. The support values and protein identifiers can be found in S1 Fig. **(B)** Comparison of the tertiary structure of Mlf homologs as predicted by AlphaFold2. Alpha-helices are shown in blue and beta-sheets in yellow, disordered regions are depicted as grey lines.

Streptophyta, and Alveolata, independent gene duplications occurred (S1 Fig), suggesting a possible functional diversification within the protein family. Structural predictions of different Mlf orthologs revealed common features in the Mlf domain, characterized by a central beta-sheet domain surrounded by alpha-helices and often flanked by disordered regions (Fig 1B). Therefore, these data suggest that Mlf is a eukaryotic protein highly conserved from unicellular to multicellular organisms, yet without a clear unifying role in eukaryotic cells.

## GiMlf associates with cellular membranes and microtubule nucleation zones

To investigate the subcellular localization of GiMlf, a C-terminal biotin acceptor peptide (BAP) or V5-tag was integrated into the endogenous gene using CRISPR/Cas9 (S2A Fig). The precise targeted recombination induced by Cas9 enabled the insertion of the tag into all four alleles of the *gimlf* gene without integrating antibiotic selection marker, use of which had previously been shown to affect *gimlf* expression in *Giardia* [25]. Furthermore, the subsequent discontinuation of selection for Cas9 expression resulted in the cell line no longer requiring antibiotic selection. Successful insertion of the BAP tag into all genomic copies was verified at both the genomic and protein levels (S2B and S2C Fig).

In line with previous reports [18,19], in vast majority of cells, GiMlf was present simultaneously in two types of cellular locations. Firstly, it was found in the vicinity of the mitosomes (Figs 2A and S3A). The GiMlf-specific signal mirrored the typical mitosomal pattern of central organelles between the two cell nuclei and peripheral organelles dispersed throughout the cytoplasm. However, the two signals did not overlap in most cases. Secondly, GiMlf was associated with specific parts of the cell cytoskeleton. In particular, thanks to the relaxed cell structure enabled by expansion microscopy, we were able to confirm its presence in the basal bodies of the flagella (Figs 2B, 2C, and S3B) and at the dense band (Figs 2B, 2C, and S3B), a structure acting as the nucleation zone for the microtubules (MTs) forming the adhesive disc of *Giardia* [26]. Importantly, GiMlf was also found at the margin of the adhesive disc (Figs 2C, 2D and S3A) [20,27], where the plus ends of MTs are located [28].

In addition, a fraction of the protein was associated with other membranes including the nuclear envelope (Fig 2C). In encysting cells, GiMlf localized at the ER cisternae, where the encystation marker and the major cyst wall component CWP1 was localized, and in the proximity of ESV membranes in the later stages of encystation (Fig 2D). Finally, the specific presence of GiMlf at the cytoskeleton and the mitosomes was confirmed by electron tomography (Figs 2E and S4).

To further analyze the association of GiMlf with cellular organelles and the cytoskeleton, the protein was detected in cellular fractions obtained from lysed *Giardia* cells (Fig 2F). GiMlf primarily appeared in the low-speed pellet (LSP) fraction, which is typically enriched in nuclei and cytoskeletal elements, aligning with its localization at the nuclear envelope, the disc margin, and the basal bodies. A smaller fraction of the protein was found in the high-speed pellet (HSP) fraction, which mainly contains mitosomes and ER. Although predicted to be a soluble protein with no transmembrane domain, GiMlf was not found in the soluble cytosolic fraction (Fig 2F).

To experimentally determine its membrane topology, the combined LSP and HSP fractions were first treated with trypsin in a protease protection assay (Fig 2G) aimed at determining whether the protein is shielded from the externally added protease within a membrane-bound compartment. GiMlf showed sensitivity to trypsin treatment (Fig 2G), with only a small amount of the protein remaining intact, indicating that it is not membrane-enclosed. Next, the pelleted LSP and HSP fractions were incubated in 100 mM sodium carbonate (pH 11), used to

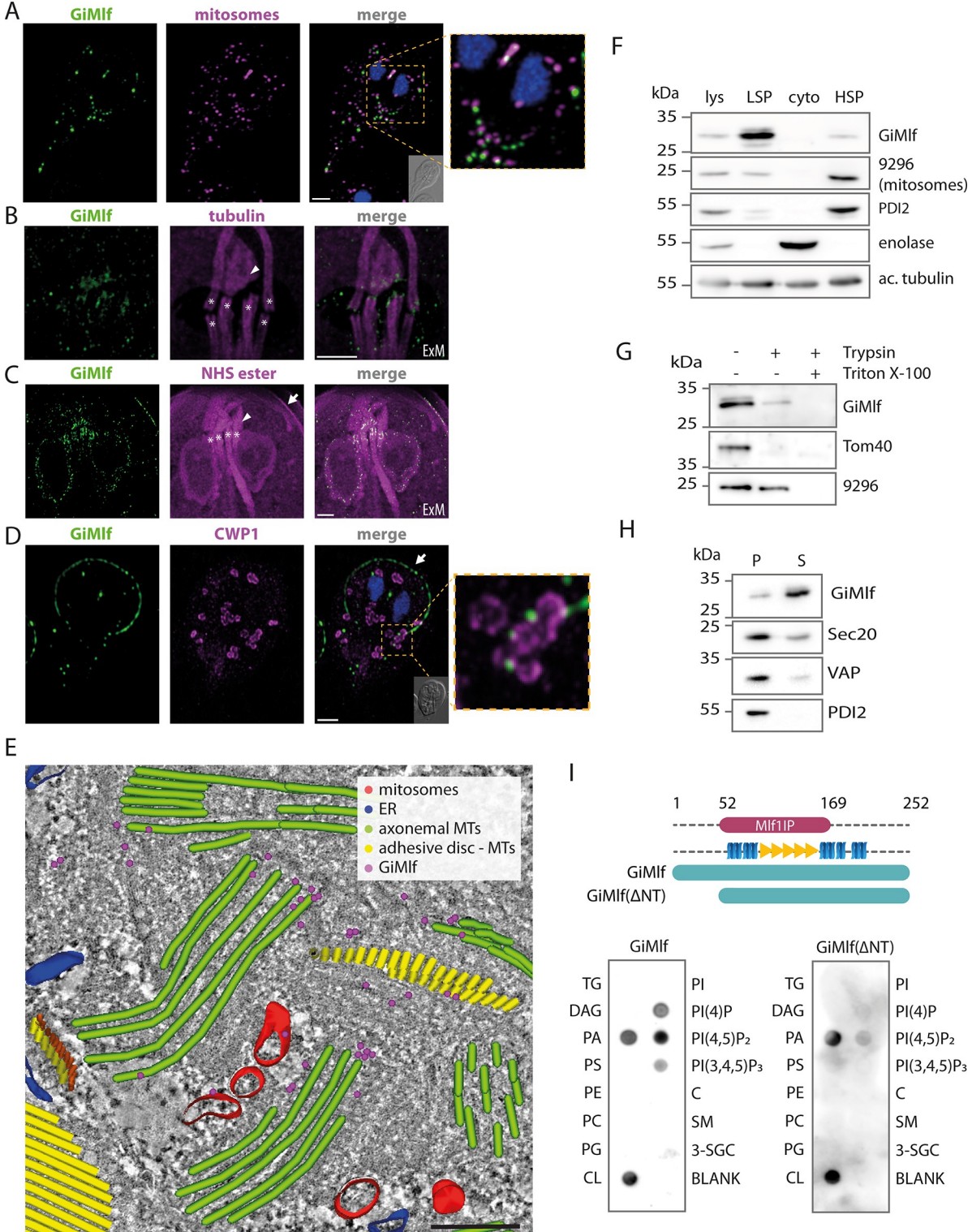

**Fig 2. GiMlf is localized in the vicinity of mitosomes, nucleation zones of cytoskeletal components, and other membrane-bound compartments. (A)** Localization of endogenously BAP-tagged GiMlf in *Giardia* using confocal microscopy. The cells were stained with an anti-BAP antibody (green), an anti-GL50803_9296 antibody (mitosomal marker; magenta), nuclei stained with DAPI (blue). DIC image of the corresponding cell is shown in the corner of the merged image, scale bar: 2 μm. **(B,C)** Localization of GiMlf in trophozoites using expansion microscopy (ExM), 3.7 expansion factor. Scale bars: 4 μm. **(B)** The cells were stained with an anti-GiMlf antibody (green) and an

anti-acetylated tubulin antibody (magenta) or **(C)** anti-BAP antibody (green) and NHS ester dye that labels primary amines of proteins. The basal bodies are marked by asterisks, the dense band is indicated by an arrowhead and the disc margin is marked by an arrow. **(D)** Localization of GiMlf in encysting cells (48 h post induction) using confocal microscopy. The cells were stained with anti-CWP1 antibody (green) and anti-BAP antibody (magenta), nuclei stained with DAPI (blue). The disc margin is marked by an arrow. The DIC image of the corresponding cell is shown in the corner of the merged image, scale bar: 2 μm. **(E)** Electron tomography of the central region between two *Giardia* nuclei depicting immunogold-labeled endogenously V5-tagged GiMlf and reconstructed subcellular structures. The image shows the presence of GiMlf (magenta dots) at the base of the basal bodies (green), adhesive disc microtubules (yellow) and at the mitosomes (red), scale bar: 250 nm. **(F)** Western blot of *Giardia* cellular fractions labeled with anti-GiMlf antibody and compartment-specific antibodies. Lysate (lys), low-speed pellet (LSP; containing cytoskeleton and nuclei), cytoplasm (cyto) and high-speed pellet (HSP; containing membrane-bound organelles), 9296—mitosomal marker protein (GL50803_9296) of unknown function, PDI2 –ER marker–protein disulfide isomerase 2, enolase–cytosolic marker, ac. tubulin–acetylated tubulin. **(G)** Trypsin treatment of the combined LSP and HSP fractions in the presence or absence of 1% Triton X-100. Tom40 and GL50803_9296 were used as markers for protease-accessible and membrane-protected proteins, respectively. **(H)** The combined fractions of LSP and HSP were treated by sodium carbonate to release peripherally associated proteins from cellular membranes. S–fraction of proteins released into the supernatant, P–fraction of proteins pelleted together with membranes. Sec20, VAP, and PDI2 were used as markers for integral membrane proteins. **(I)** The membrane lipid strip assay demonstrates the affinity of GiMlf for signaling components within the membrane. Recombinant GiMlf was detected using an anti-GiMlf antibody. TG–triglyceride, DAG–diacylglycerol, PA–phosphatidic acid, PS–phosphatidylserine, PE–phosphatidylethanolamine, PC–phosphatidylcholine, PG–phosphatidylglycerol, CL–cardiolipin, PI–phosphatidylinositol, C–cholesterol, SM–sphingomyelin, 3-SGC–3-sulfogalactosylceramide, PI(4)P–phosphatidylinositol (4)-phosphate, PI(4,5)P$_2$ –phosphatidylinositol (4,5)-bisphosphate, PI(3,4,5)P$_3$ – phosphatidylinositol (3,4,5)-trisphosphate.

separate peripherally associated proteins from cellular membranes. Unlike integral membrane proteins (Sec20, VAP and PDI2), GiMlf was found in the supernatant after the treatment (Fig 2H), suggesting that it associates peripherally with cellular membranes without integrating into the lipid bilayer.

## Recombinant GiMlf binds signaling phospholipids

To understand the basis for the association of GiMlf with cellular membranes, we experimentally tested whether it possesses some lipid-binding properties. For this purpose, recombinant GiMlf was purified from *E. coli* (S5B Fig), and applied to a membrane lipid strip with a range of structural and signaling lipids. Using a specific polyclonal antibody raised against GiMlf, a selective binding to signaling lipids, such as phosphatidic acid (PA), phosphatidylinositol phosphates (PIPs), and cardiolipin was revealed (Fig 2I). The interaction with PIPs appears to be tied to its N-terminal disordered region, as evidenced by the significantly reduced binding properties of the N-terminally truncated variant lacking the first 53 amino acids (GiMlf-ΔNT) (Figs 2I and S5). Together, these findings indicate that GiMlf is recruited to specific cellular membranes by recognizing signaling phospholipids and that the N- terminal region might confer specificity towards different types of these lipids.

## GiMlf interacts with Hsp40 and a proteasome subunit

Given the range of possible functions GiMlf could play at the different cellular locations, we isolated its putative interaction partners using a biotin affinity purification assay with chemical crosslinking [18]. To this end, the gene encoding the *E. coli* biotin ligase, BirA, was introduced into the cell line with endogenously BAP-tagged GiMlf. When expressed together in the same cellular compartment, BirA mediates specific biotinylation of a protein carrying the BAP peptide (S6A Fig).

Using streptavidin-coated magnetic beads, the *in vivo* biotinylated GiMlf and its putative interacting partners were isolated from the combined LSP and HSP fractions after chemical crosslinking with DSP (S6B Fig). MS analysis of the eluted sample, performed in biological and technical triplicates, identified 49 significantly enriched proteins (P-value <0.05, fold change >2) compared with the control sample (Fig 3A, full list of proteins in S1 Table). GiMlf was identified as the most abundant protein, validating the specificity of the isolation

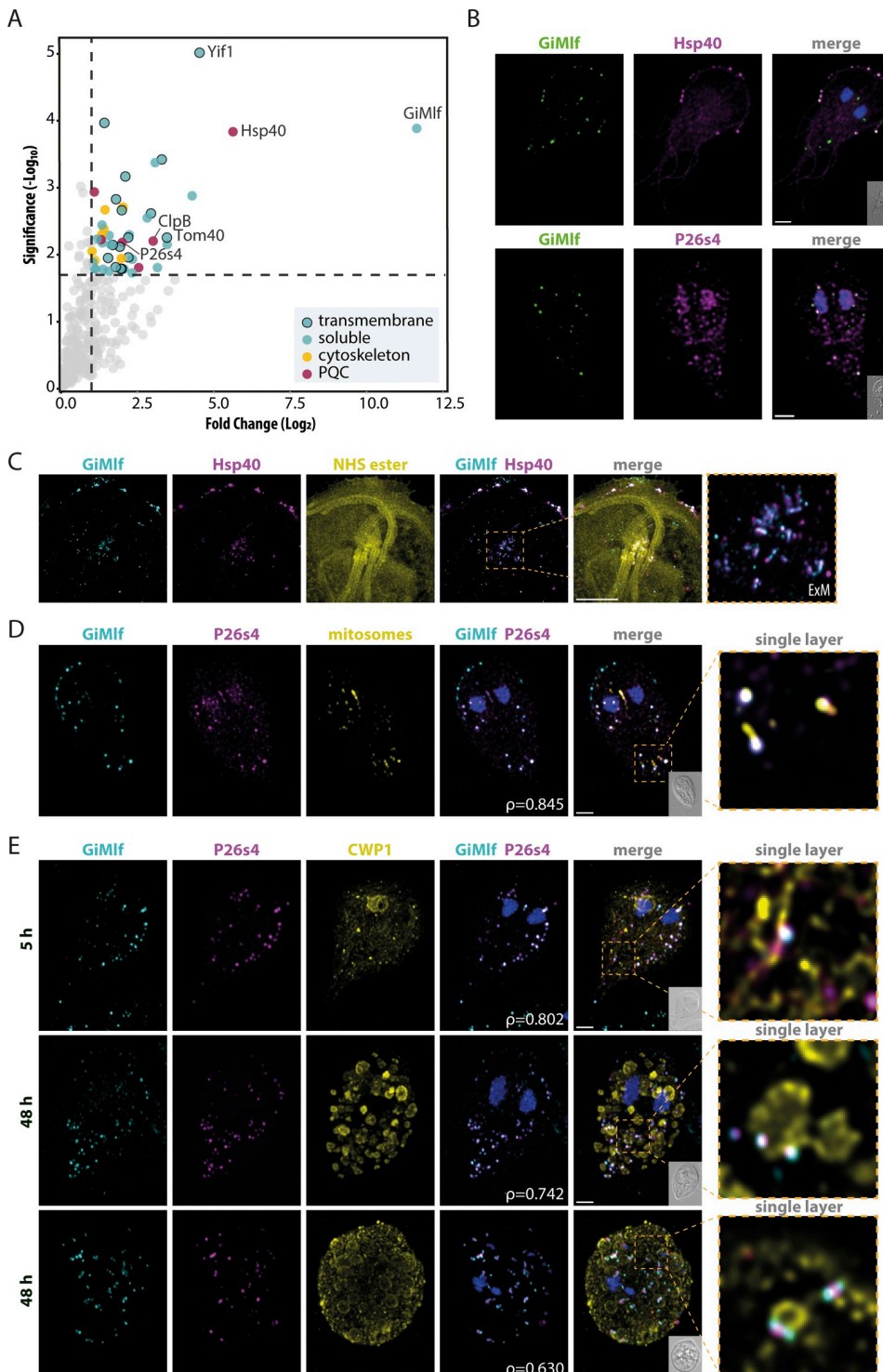

**Fig 3. The interactome of GiMlf includes transmembrane proteins, cytoskeletal components, chaperones, and a proteasomal subunit. (A)** Volcano plot of the isolated GiMlf interactome using DSP crosslinking (P≤0.02, fold change >2, n = 3). Cytoskeletal components, PQC proteins, and transmembrane proteins are highlighted as indicated in the legend. Transmembrane topology prediction was performed using the DeepTMHMM tool. The indicated proteins were further analyzed in this study. **(B)** Localization of endogenously BAP-tagged GiMlf and V5-tagged putative interacting partners in trophozoites of *G. intestinalis* using confocal microscopy. The cells were stained with anti-BAP antibody (green) and anti-V5 antibody (magenta), scale bar: 2 μm **(C)** Localization of endogenously BAP-tagged GiMlf and V5-tagged Hsp40 at the basal bodies in trophozoites using expansion microscopy (ExM), scale bar:

10 µm. The cells were stained with anti-BAP antibody (cyan) and anti-V5 antibody (magenta), and NHS ester (primary amines of proteins, yellow). **(D)** Localization of endogenously BAP-tagged GiMlf and V5-tagged P26s4 at mitosomes in trophozoites of *G. intestinalis* using confocal microscopy. The cells were stained with anti-BAP antibody (cyan) and anti-V5 antibody (magenta), and an anti-GL50803_9296 antibody (yellow, mitosomal marker), scale bar: 2 µm The enlarged image shows a single layer of the image stack. Pearson's correlation coefficient ($\rho$ = 0.8448) was calculated for the subsection of GiMlf and P26s4 that colocalize in proximity to mitosomes. **(E)** Localization of endogenously BAP-tagged GiMlf and V5-tagged P26s4 in encysting cells of *G. intestinalis* in the early (5 hpi) and late (48 hpi) phases of encystation using confocal microscopy. The cells were stained with anti-BAP antibody (cyan) and anti-V5 antibody (magenta), and anti-CWP1 antibody (yellow), scale bar: 2 µm. The enlarged images show a single layer of the image stack. Pearson's correlation coefficient ($\rho$) was calculated for each stack. Top panel—$\rho$ = 0.802, middle panel—$\rho$ = 0.742, bottom panel—$\rho$ = 0.630.

procedure (Fig 3A). Approximately one-third of the isolated proteins had transmembrane domains (Fig 3A), corresponding to the association of GiMlf with subcellular membranes (Fig 2).

The putative interaction partners included 17 proteins with experimentally confirmed localization in *Giardia* (S1 Table), mostly at sites corresponding to the identified GiMlf localization, such as mitosomes (Tom40, GL50803_17276, GL50803_7188, and Grx5) (S6C Fig), with the first two representing outer mitosomal membrane proteins ([18,19,29,30], ER (nucleoside transporter, dolichyl-diphosphooligosaccharide-protein glycotransferase and BiP) [24,31,32], basal bodies and the cytoskeleton (GL50803_6709, GL50803_29796 and GL50803_15445, glyceraldehyde-3-phosphate dehydrogenase, Flagella-associated protein, and glutamate dehydrogenase) [30,33,34], and the cytosol (Hsp70, alcohol dehydrogenase, and kinases GL50803_16824 and GL50803_13215) [35].

Of the remaining set of proteins, four hits carrying known protein domains were selected for further analysis. These included the chaperone Hsp40 (GL50803_17483), the protein disaggregase ClpB/Skd3 (GL50803_17520), the vesicular trafficking protein Yip1 interacting factor (Yif1, GL50803_12945), and the regulatory subunit of the 26S proteasome P26s4/Rpt2 (GL50803_113554). To assess their colocalization with GiMlf, plasmids carrying the corresponding genes with C-terminal V5 tag were introduced into the cell line with BAP-tagged GiMlf. Two of the selected proteins, Hsp40 protein and P26s4 indeed colocalized with GiMlf, yet at two separate locations in the cell (Fig 3B).

Hsp40 was either present at the axonemes (52% of observed cells, n = 50) (S6D Fig) or displayed ubiquitous cytoplasmic localization, appearing also at the flagella (48% of observed cells) and in 40% of cells it was additionally present at the disc margin, where it colocalized with GiMlf (Fig 3B). A more detailed insight was provided by expansion microscopy that revealed the proteins also colocalized at the basal bodies of the flagella and the dense band of the adhesive disc (Fig 3C), confirming their presence at the central sites of the cytoskeleton, which are hotspots for protein folding and assembly.

P26s4 displayed nuclear and cytosolic localization (Fig 3B) that corresponds to the known distribution of proteasomes in *Giardia* [36]. In half of the observed cells (n = 50), it displayed specific localization in cytosolic foci (Fig 3D), where it colocalized with GiMlf. Interestingly, co-staining with a mitosomal marker revealed that the co-localization of GiMlf and P26s4 occurs in the vicinity of mitosomes in the forementioned foci (Fig 3D). P26s4 was previously shown to be responsible for the compartment-specific recruitment of the 26S proteasome [37] and this data suggested that together with GiMlf they may be involved in protein quality control (PQC) near the mitosomal surface.

Proteasome activity increases during *Giardia* encystation [38], a process that involves an overall remodelling of membrane-bound compartments and the cytoskeleton. Thus, we

followed the distribution of P26s4 and GiMlf during the encystation. In its early stages (5 hours post induction–hpi) P26s4 colocalized with GiMlf at defined points of the ER, where CWP1 synthesis was taking place (Fig 3E). As the encystation progressed and the ESVs were formed (48 hpi), P26s4 remained with GiMlf at single foci at the membrane of the maturing vesicles. Interestingly, the position of the two proteins changed towards the late stage of encystation, when the cell typically becomes rounded, the nuclei undergo mitosis, and the CWP1 protein is sorted towards the periphery of the ESVs, forming ring-like structures [39]. At this stage, P26s4 and GiMlf separated to form a uniform pattern across the cell in which P26s4 was surrounded by two GiMlf signals but still remained associated with the ESVs (Fig 3E).

The other two selected putative interacting proteins, Yif1 and ClpB homologs, showed no clear colocalization with GiMlf under physiological conditions (S6E Fig). Both proteins were found in discrete puncti throughout the cytosol and, in the case of ClpB, also in the nuclei.

## GiMlf is upregulated in response to cellular stress, relocalizes to affected sites, and accumulates in cytosolic foci

To further understand the role of GiMlf in PQC, we designed additional experiments. The association of GiMlf with organellar membranes prompted us to induce compartment-specific proteotoxic stress by overexpressing specific membrane proteins. To this end, we monitored the cellular localization and protein levels of GiMlf in cell lines overexpressing the ER or mitosomal transmembrane proteins, Get2 and Tom40, respectively. Western blot analysis revealed a significant increase in the expression of GiMlf, by 2-fold and 2.1-fold respectively, in the transfected cell lines compared with the control (Figs 4A and S7A). Notably, GiMlf relocated to the affected compartments, specifically to the ER membrane with Get2 overexpression and to the mitosomal membrane with Tom40 overexpression (Fig 4B).

Furthermore, we exposed *Giardia* to a 43˚C heat shock, as such conditions have been previously shown to elicit a heat shock response in *Giardia* [40]. The expression of GiMlf was monitored using western blotting after a 20-minute incubation at the elevated temperature, followed by a recovery period of up to 30 hours at 37˚C (Fig 4C). Additionally, expression of the molecular chaperone BiP was followed as a control to verify the anticipated heat shock response. While BiP levels increased sharply immediately after the heat shock, the increase in GiMlf protein levels was more gradual, reaching a peak at 4 hpi. Both proteins returned to normal levels prior to 16 hpi (Fig 4C).

The rise in GiMlf protein levels is consistent with previous transcriptomic analyses [23,24] that showed upregulation of GiMlf following a 40˚C heat shock, along with two Hsp70 proteins (GL50803_16412, GL50803_88765) identified as putative interaction partners of GiMlf in the present study (S1 Table).

The upregulation of GiMlf was accompanied by dynamic changes in its cellular distribution (Fig 4D). Within 30 minutes of recovery, an increased GiMlf signal was apparent in 84% of the observed cells. Most cells still displayed the presence of GiMlf foci but also a widespread cytoplasmic distribution (transitional state I). In approximately 30% of the cells, widespread GiMlf cytoplasmic staining became dominant (ubiquitous). Within 2 hours of recovery, the proportion of cells exhibiting widespread cytoplasmic localization of GiMlf increased to 58%. Interestingly, after approximately 1 hour, cells presenting numerous GiMlf-positive foci, the majority of which were again near mitosomes, began to emerge (4%) (transitional state II), and by 6 hours, this became the predominant phenotype, observed in 62% of the cells (abundant foci). Subsequently, the proportion of cells with the steady state phenotype started to increase, reflecting the reduction in GiMlf protein levels. These observations strongly imply that GiMlf is instrumental in cellular recovery from proteotoxic stress, which involves the

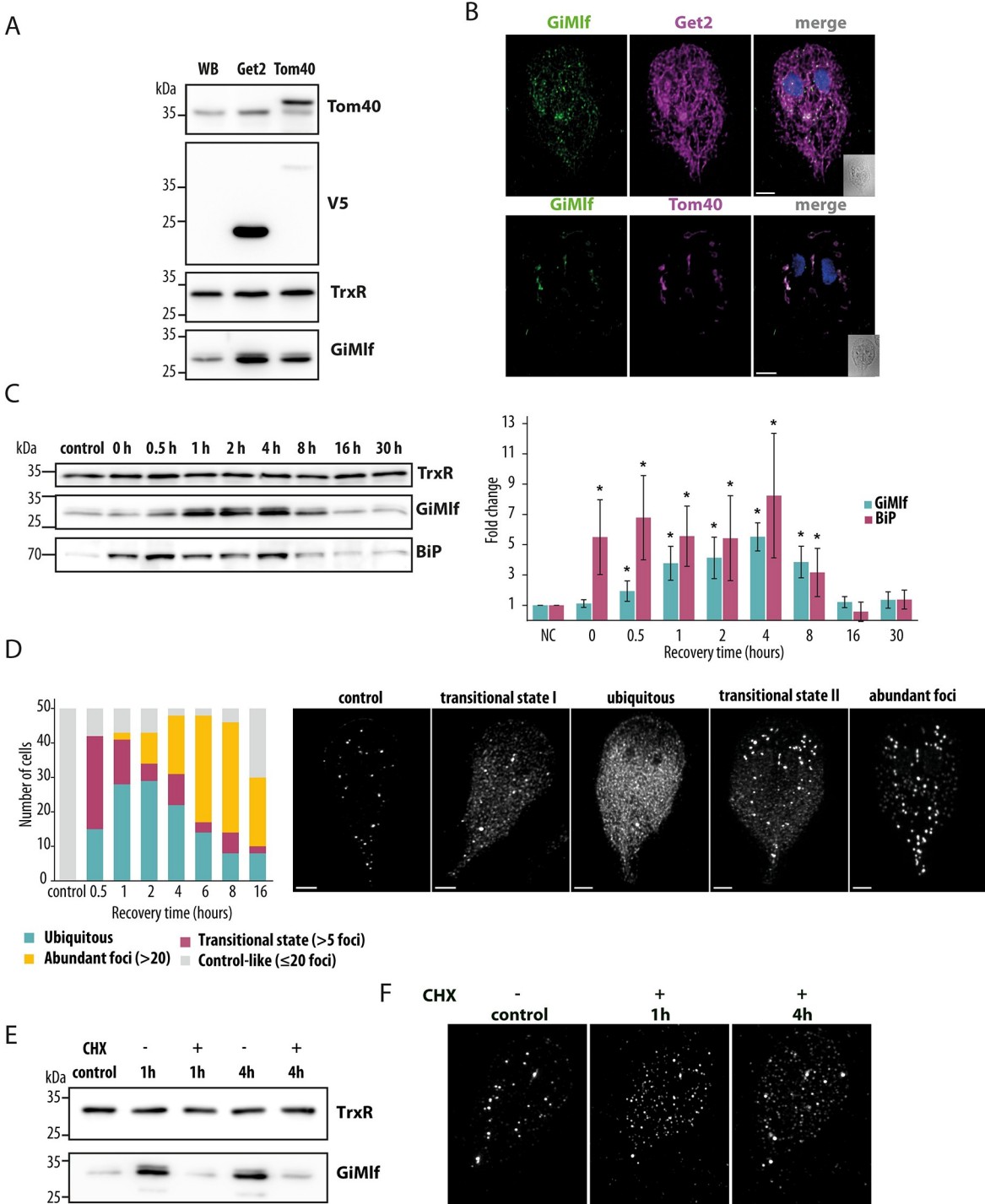

**Fig 4. GiMlf responses to proteotoxic stress by upregulation and relocalization. (A)** Comparison of protein levels of GiMlf detected by western blot in the wild-type (WB) cell line and in cell lines episomally expressing either Get2-V5 (ER) or Tom40-V5 (mitosome). The cell lysates were probed with indicated antibodies. Thioredoxin reductase (TrxR) served as a loading control. **(B)** Localization of GiMlf and V5-tagged overexpressed membrane proteins in *G. intestinalis* using confocal microscopy. The cells were stained with an anti-GiMlf antibody (green) and an anti-V5 antibody (magenta), DNA was stained with DAPI (blue), scale bars: 2 μm **(C)** Comparison of protein levels of GiMlf and BiP following a 20-minute heat shock (43˚C) and an indicated recovery period (37˚C), as detected by western blot. The cell lysates were labeled with the indicated antibodies (one of three experiments is shown). TrxR served as a control. The graph shows the changes in the protein levels of GiMlf (cyan) and BiP (magenta) after the heat shock and the indicated recovery periods, relative to the control (n = 3). Statistically significant changes in protein levels (P-value < 0.05, one-tailed t-test) are highlighted with asterisks. **(D)** Left–

Plotted changes in GiMlf distribution in individual cells (n = 50) during recovery after heat shock. Right–Examples of different phenotypic distributions of GiMlf observed by confocal microscopy. The cells were stained with anti-BAP antibody, scale bars: 2 μm. **(E)** Protein levels of GiMlf 1h and 4h after heat shock in the presence/absence of cycloheximide (CHX). Control–untreated cells. The cell lysates were probed with anti-TrxR and anti-BAP antibodies. TrxR served as a loading control. **(F)** Localization of GiMlf in *G. intestinalis* using confocal microscopy during recovery phase after heat shock in CHX treated cells. Control–untreated cells, steady state. The cells were stained with anti-BAP antibody, scale bars: 2 μm.

accumulation of protein aggregates and misfolded or denatured proteins. Finally, we tested if the ubiquitously localized GiMlf in the recovery phase after the heat shock comprises of only newly expressed protein or if the existing protein can redistribute from the foci. To this end, the cells were treated with cycloheximide (CHX) to effectively block protein translation (Fig 4E) and then exposed to a 43°C heat shock. After 1h of recovery time, GiMlf partially dispersed from the foci into smaller bodies across the cytoplasm (Fig 4F andS7B), suggesting that the protein can re-distribute from the foci. Later, the protein appeared to concentrate more in the foci, although not as prominently as in the control cells. The effect of CHX was manifested by the absence of the widespread cytoplasmic localization of GiMlf, which likely corresponds to newly translated protein induced under the stress conditions.

## GiMlf-P26s4 foci are ubiquitin-rich bodies that become enlarged in the heat shock recovery phase

Several assumptions prompted us to test the possible presence of ubiquitinylated proteins in GiMlf foci. First, the interaction with a proteasome subunit suggests the involvement of ubiquitin (Ub)-labeled substrates targeted for degradation [41]. Second, Hsp40 proteins are known to condensate into Ub-rich membraneless organelles to mitigate protein aggregation [42]. Third, in an animal model, the Mlf2 paralogue was localized to nuclear Ub-rich foci in cells with nuclear pore assembly defect [8]. Labeling cells with an anti-Ub antibody revealed distinct Ub locations throughout the cytoplasm of *Giardia*, identifying several Ub-rich foci within the cytoplasm and at the disc margin. Indeed, these cytoplasmic foci overlapped with the GiMlf signal, and the Ub-specific signal at the disc margin exhibited partial co-localization with GiMlf (Fig 5A). Following a 4-hour recovery from heat shock, the presence of Ub-rich foci containing GiMlf became more prominent (Fig 5A). These observations demonstrated that cytoplasmic foci containing GiMlf are Ub-rich bodies and that ubiquitinylation also appears to play a role in the protein turnover at the adhesive disc margin.

To obtain further insight into the structure of GiMlf foci, we monitored their morphology at steady state and in the heat shock recovery phase using stimulated emission depletion (STED) microscopy. At steady state, GiMlf formed mainly filled punctate structures of 182 nm in diameter in average (n = 30, Fig 5B). In the recovery phase, donut-shaped hollow structures were observed. Their size significantly increased to an average of 279 nm (n = 30, P-value = 1.72E-09) and adopted either a circular or ellipsoidal shape, often in a conjoined arrangement (Fig 5B). When the cells were co-labeled for P26s4, two distinct scenarios could be observed in the heat shock recovery phase. P26s4 was either found on the edge of the foci together with GiMlf or was excluded to the periphery of the GiMlf signal (Fig 5C). It is difficult to determine the time sequence of the two stages without live cell microscopy, but the latter scenario corresponded with the confocal images showing reduced co-localization of the two proteins at later stages of recovery (S7C Fig) or *Giardia* encystation (Fig 3E).

It is plausible that these foci are in fact built by the oligomerized GiMlf subunits as high molecular weight species corresponding to GiMlf octamers, as observed by size exclusion chromatography of purified recombinant GiMlf lacking the C-terminal disordered region (ΔCT) because of its much better solubility (S8 Fig).

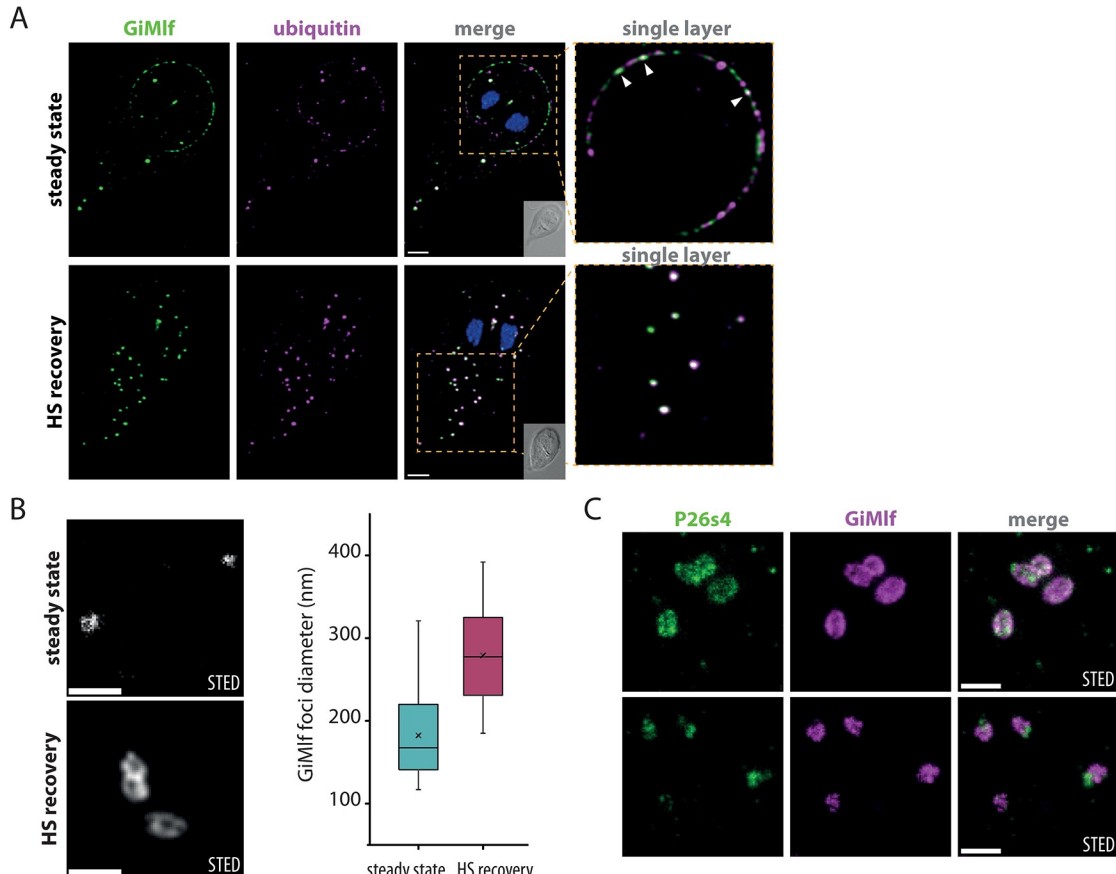

**Fig 5. GiMlf-P26S4 foci are ubiquitin-rich bodies that enlarge after heat shock. (A)** Localization of GiMlf and ubiquitin in *G. intestinalis* using confocal microscopy in steady state and after 4 hours of recovery after heat shock. The cells were stained with anti-BAP (green) and anti-ubiquitin (magenta) antibodies. The enlarged images show a single layer of the image stack. The arrows point to locations on the disc margin, where ubiquitin and GiMlf colocalize. Nucleic DNA was stained with DAPI (blue). DIC image of the corresponding cell is shown in the corner of the merged image. Scale bars: 2 μm. **(B)** Comparison of GiMlf foci before and after heat shock and 4 hours of recovery using STED microscopy. The cells were stained with anti-BAP antibody. Scale bars: 0.5 μm. Box plots compare the foci diameters measured at its widest point in each state (n = 30; significant difference determined with the two-tailed t-test with equal variance, P-value = $1.72 \times E^{-9}$). **(C)** Localization of GiMlf and V5-tagged P26s4 after heat shock and 4 hours of recovery using STED microscopy. The cells were stained with anti-BAP (green) and anti-V5 (magenta) antibodies. Scale bars: 0.5 μm.

### The absence of GiMlf results in the mislocalization of Hsp40, reduction in P26s4 foci, and broad changes in proteostasis

The above experiments indicated that GiMlf could be a cellular factor responsive to proteotoxic stress, involved in the formation of cellular sites for protein folding via its interaction with Hsp40 or protein degradation through its binding to P26s4. Therefore, we decided to analyze the phenotype of cells lacking GiMlf, with a specific focus on its interacting factors and overall proteostasis. To accomplish this, we used a cell line devoid of all four alleles of the *gimlf* gene (ΔGiMlf) that was initially developed as a proof of concept during the implementation of the CRISPR/Cas9 knockout methodology in *Giardia* [43]. The ΔGiMlf cell line, while viable, exhibited significant growth impairment (S9A Fig, P-value of 0.024). Morphological analysis of the ΔGiMlf cells did not reveal any prominent cellular defects (S9B Fig). Analysis of the encysting ΔGiMlf cells showed successful formation of ESVs and subsequent release of CWP1 onto the cyst surface (S9C Fig). However, there was an approximately twofold increase in the

total number of cysts in the ΔGiMlf cell population (Fig 6A), suggesting that the absence of GiMlf steers the cells to encystation. However, the underlying mechanisms that lead to this outcome remain unclear and require further investigation.

Next, we analyzed the impact of GiMlf absence on the cellular distribution of its interacting partners P26s4 and Hsp40. For this purpose, V5-tagged P26s4 or Hsp40 were expressed in ΔGiMlf cell line. The amounts of the V5-tagged proteins were comparable to those in control cells (S9D Fig), but their cellular distribution was significantly altered in the ΔGiMlf cell line. Prominent changes were observed in the distribution of Hsp40, which showed an increase in the percentage of cells with widespread cytoplasmic localization from 8% to 44% (n = 50), whereas its presence was diminished at the disc margin (Fig 6B). There was also a significant decrease in the number of the P26s4 foci that neighbor mitosomes as well as their average maximum intensity in the ΔGiMlf cell line (Fig 6C).

When ΔGiMlf cells were exposed to heat shock and subsequent four-hour recovery period, there was no noticeable change in the P26s4 phenotype compared with untreated cells (Fig 6C). However, in the case of Hsp40, after heat shock and recovery, the protein relocalized from the cytoplasm to the axonemes and disc margin, where it did not form distinct foci as in the control cells (Fig 6B). In addition, in 40% of the observed cells (n = 50), Hsp40 was present at the flagellar tips in the ΔGiMlf cell line (Fig 6B), phenotype that was rarely observed in the control cells. In conclusion, the absence of GiMlf affected the cellular localization of its interacting partners, bolstering a functional model in which GiMlf participates in the accumulation of P26s4 and Hsp40 at the cellular sites of their action.

To gain a deeper understanding of how the absence of GiMlf affects the entire cellular proteome, we conducted a proteomic analysis of the ΔGiMlf cells. The analysis was expanded by including cells overexpressing GiMlf (GiMlf-OE). Notably, these cells did not exhibit any growth- or encystation-related phenotypes (Figs S9A and 6A), but the protein overexpression led to the formation of large vesicular structures containing large deposits of GiMlf (S10A and S10B Fig) [18,44].

The protein profiles of both ΔGiMlf and GiMlf-OE cell lines were analyzed using semi-quantitative label-free proteomics alongside the corresponding controls. In total, 192 proteins were significantly upregulated and 307 were significantly downregulated in the ΔGiMlf cell line, as determined by a two-sample test of the biological triplicate analysis (permutation-based FDR set at 0.05; full list of detected proteins in S2 Table), with 332 categorized into distinct functional groups (Fig 6D).

As indicated by the phenotypic analysis, the most notable changes included processes that maintain proteostasis (Fig 6D and S2 Table). We observed an upregulation of protein synthesis and ribosome biogenesis. Conversely, there was a marked downregulation of molecular chaperones, V-ATPase, proteins involved in vesicular transport and fusion, as well as substantial changes in the levels of the (de)phosphorylating enzymes, indicating changes in cellular signaling. Interestingly, these changes were accompanied by the downregulation of glycolytic enzymes, especially glyceraldehyde 3-phosphate dehydrogenase and phosphoglycerate kinase (S2 Table). The overall changes in cell surface antigens correspond to the natural antigenic variability of the parasite [45].

Interestingly, in contrast to the knockout cells, proteomic analysis of the GiMlf-OE cell line showed only limited changes. Most of the 11 upregulated and 6 downregulated proteins involved surface antigens (S3 Table), and no significant changes in other cellular processes were detected.

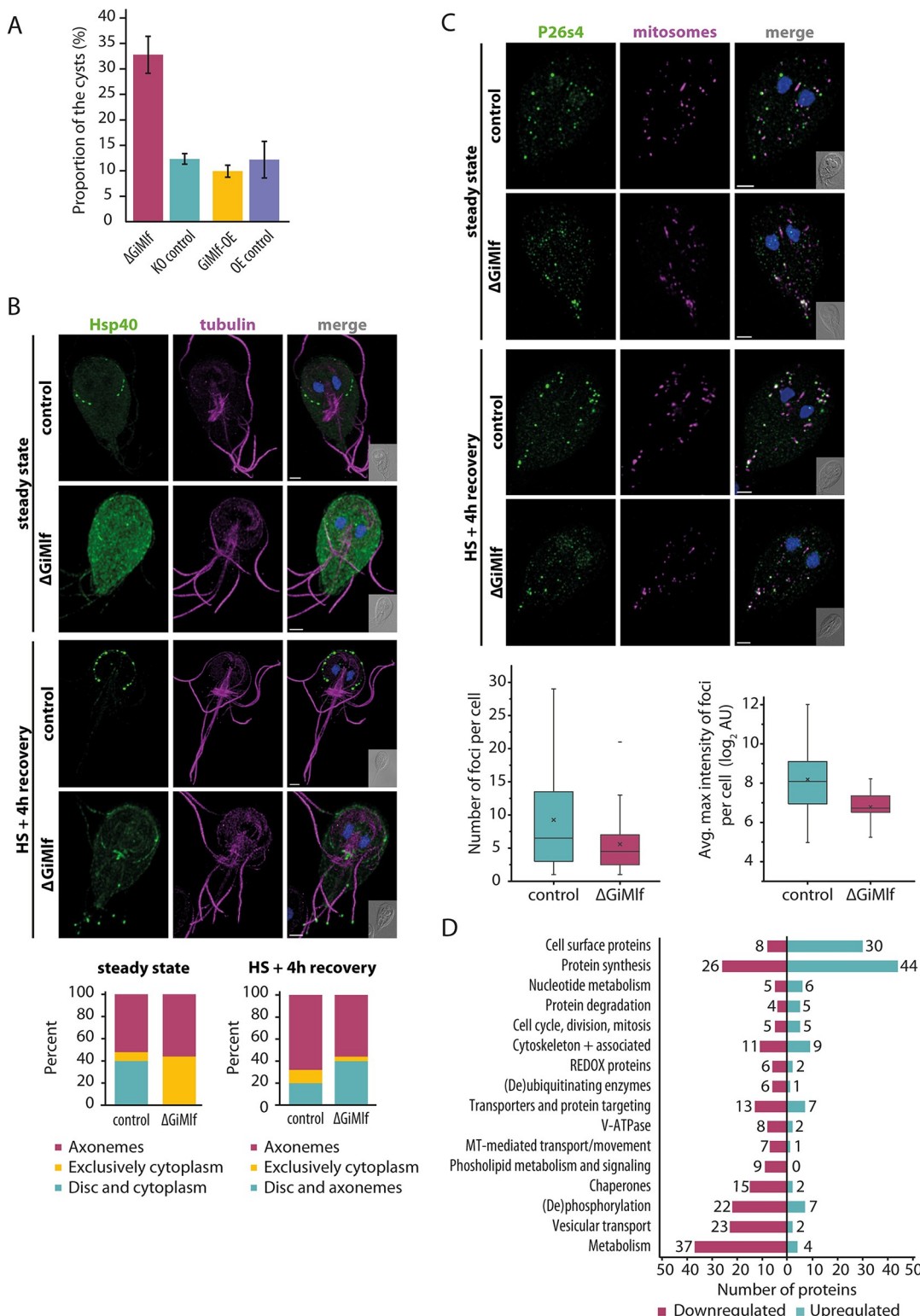

**Fig 6. The absence of GiMlf leads to altered localization of Hsp40, a reduction in P26s4 foci, and broad changes in proteostasis. (A)** Comparison of cyst production of ΔGiMlf cell line, GiMlf-OE cell line and their respective control cell lines after 48 h of encystation (n = 3). (Two-tailed t-test with equal variance, P-value = 0.002). The cyst production of GiMlf-V5-overexpression cell line did not differ significantly (Two-tailed t-test with equal variance, P-value = 0.45) when compared to its control. **(B)** Top–Localization of V5-tagged Hsp40 in control and ΔGiMlf cells in steady state and after 4 hours of

recovery after heat shock. The cells were stained with an anti-V5 antibody (green) and an anti-acetylated tubulin (magenta). Nucleic DNA was stained with DAPI (blue). Scale bars: 2 μm. Bottom–The plotted overall distribution of Hsp40 in steady state and at 4 hours of recovery after heat shock (n = 50). **(C)** Top–Localization of V5-tagged P26S4 in control and ΔGiMlf cells under steady conditions and after 4 hours of recovery after heat shock. The cells were stained with an anti-V5 antibody (green) and an anti-GL50803_9296 antibody (mitosomal marker; magenta). Nucleic DNA was stained with DAPI (blue). Scale bars: 2 μm. Bottom–The box plots compare the number (left; significance determined with two-tailed t-test with unequal variance, P-value = 0.01) and the average max intensity (right; significance determined with two-tailed t-test with unequal variance, P-value = 0.002) in control and ΔGiMlf cells in steady state (n = 40). **(D)** A diagram depicting number of proteins with significant changes in expression within different functional groups in the ΔGiMlf cell line. Statistical significance was assessed using a two-sample test with permutation-based FDR with a value set at 0.05 (n = 3).

## Discussion

Human Mlf1 was originally identified as part of the leukemic fusion protein NPM-MLF1, generated by chromosomal translocation in patients with acute myeloid leukemia [1]. Since its discovery, Mlf1 has been linked to a variety of cellular functions, however, only little explanation has been provided for its mechanism of action [2]. In this study, we demonstrate that Mlf1 is widespread in most of the major eukaryotic supergroups, including the single-cell organisms, suggesting that the protein was likely present in LECA. In fact, it has been lost in very few lineages and has even been preserved in parasitic organisms like *Giardia* with a highly reduced genome and simplified organellar system [9].

By studying the Mlf1 homolog in *Giardia* (GiMlf), we have shown that the protein localizes to distinct functional foci involved in protein folding and degradation. In *Giardia*, these distinct sites are marked by different interaction partners, Hsp40 at the cytoskeletal structures, and proteasome subunit P26s4 in the vicinity of mitochondrial organelles known as mitosomes [46]. Importantly, we showed that GiMlf-specific foci overlap with Ub-rich bodies, which are typically formed to prevent protein aggregation [42] or concentrate proteasomes for targeted degradation [47]. Following proteotoxic stress, we observed an upregulation of GiMlf expression, leading to its expansion from the foci into a broad cytoplasmic distribution before it reassembled into newly formed, enlarged foci. At this juncture, GiMlf foci not only increase in size but also organize into 'hollow' protein structures devoid of GiMlf at their core. This pattern suggests the formation of membraneless organelles, which is characteristic of the induction of aggresomes or proteasome assemblies [48].

Central to *in vivo* protein phase separation is the role of intrinsically disordered regions (IDRs) [49]. GiMlf, along with its homologs, typically possesses the central Mlf domain, which is flanked by N-terminal and C-terminal IDRs, which may drive the formation of such functionally specialized condensates. Reflecting on this function, the absence of GiMlf leads to the disappearance of well-defined Hsp40 foci, resulting in either a diffuse cytoplasmic distribution or its concentration at the flagellar tips after heat shock. Similarly, in the absence of GiMlf, P26s4 foci become smaller with reduced P26s4 content, as implied by their decreased fluorescence intensity, strongly suggesting that GiMlf participates in the formation of functionally specialized foci.

In fact, some of these functional aspects of GiMlf have been independently observed in human and animal cells. Patients with mutated Hsp40 (DNAJB6) were found to accumulate Hsp40 in clusters with Mlf1 [6], and mice with experimental overexpression of Mlf1 contained Hsp40 in the formed protein aggregates [50]. The Mlf2 paralogue has been found in Ub-rich foci in nuclear blebs formed in Torsin ATPase mutants of HeLa cells [8]. The protein aggregates formed in Huntington's disease were shown to contain Mlf1, but overexpression of Mlf1 or Mlf2 leads to the release of proteins from these aggregates [51].

Interestingly, localization of GiMlf to basal bodies of *Giardia* flagella and the dense band of the adhesive disc aligns well with the identified involvement of Mlf1 in human ciliogenesis.

Mlf1 was first localized to human cilia [52,53] and was later identified as a regulator of ciliogenesis in a genetic screen [5]. In the *Giardia* model, GiMlf responds to compartment-specific stress introduced by membrane protein overexpression, as demonstrated by the overexpression of Get2 and Tom40. *In vitro* experiments suggest that this recruitment of GiMlf may be mediated by its capacity to bind to signaling lipids such as PA, CL, and PIPs. However, we were unable to support recent findings on the role of GiMlf in autophagy and its positive impact on encystation [21]. We did not observe the formation of membrane-bound GiMlf vesicles or identify putative autophagy factors as interacting proteins. Furthermore, we noted a significant increase in cyst production in the GiMlf knockout cell line, which indicates a further downstream impact of the protein absence. Whether it relates to the downregulation of glycolysis and connected metabolic pathways, by which glucose may be preserved for the synthesis of N-acetyl galactosamine polymer, the sugar component of the cyst wall [54], calls for further investigation. The colocalization of GiMlf and the proteasome subunit at ESVs corroborates a previous study on the possible role of proteasome in the biogenesis and maturation of these secretory vesicles [17]. The precise role of proteasomes in this process remains unknown, as inhibition of proteasomes does not reduce cyst production but reduces their viability [55].

Finally, our study showcases the methodological utility of the CRISPR/Cas9 system in *Giardia*. The efficiency of homologous recombination of donor sequences into all four alleles of the *Giardia* genome induced by CRISPR/Cas9 circumvents the need for a selection marker in the recombination cassette. This technique also offers an approach for inserting multiple tags into different genes within the same cell line, a particularly valuable approach given the limited repertoire of selection markers available in *Giardia*. Furthermore, this strategy minimizes the potential adverse effects of selection antibiotics on cells, and in cases like ours, on the studied phenotypes.

## Methods and materials

### Bioinformatic analyses

Sequences for the phylogenetic analysis have been aligned using the MAFFT version 7 with the L-INS-i parameter and 1000 repetitions of iterative refinement [56]. The alignment has been automatically trimmed with the TrimAl tool [57]; positions that contained more than 30% gaps were removed. IQ-TREE version 1.6.12 has been deployed for the phylogenetic analysis, using the LG protein substitution matrix with discrete Gamma (LG+G4) [58]. The alignment visualization has been done with the ESPript version 3.0 [59]. The structural model of Mlf proteins has been predicted using AlphaFold2 [60] with the ptm model and custom MSA (GiMlf) or default settings (*H. sapiens*, *A. thaliana*, *P. marinus* Mlf proteins).

### Cell culture, strains and subcloning and encystation

Trophozoites of *G. intestinalis* (strain WB, ATCC 30957) were cultivated at 37˚C under anaerobic conditions in TYI-S-33 medium [61] supplemented with 0.1% bile (isolated from bovine gall bladder), 10% heat-inactivated adult bovine serum (PAA laboratories) and 1% penicillin-streptomycin antibiotic (Sigma-Aldrich). Strains with the pONDRA plasmid [62] (inserts: Cas9, P26s4-V5, Hsp40-V5, Yif1-V5, ClpB-V5, BirA) were selected with G418 antibiotic (600 μg/ml), strains with pTGUIDE [43], and strains with pTG plasmids [63] with inserts: V5-Get2 [64], V5-Tom40 and the GiMlf-KO cell line [43] were selected with puromycin (57 μg/ml). To subclone the Cas9-GiMlfendoBAP strain, the culture grown to full confluence was placed on ice for 10 min to detach the cells, after which the cells were counted using Beckman Coulter Z2 cell counter and diluted to a concentration 500 cells/ml. 2μl drops of the diluted cells were placed in individual wells 96-well cell culture plate (VWR) and checked

under the microscope. The wells containing a single cell were filled with 200 μl of the TYI-S33 medium and the plates were grown under anaerobic conditions at 37˚C. When the cells reached full confluence, they were transferred to standard cultivation tubes. The genomic DNA was tested by PCR for the presence of the integrated tag using primers BAP-F and GiMlf-3UTR(1000bp)-PacI-R, the presence of the native form of the gene with primers GiMlf-BAP-rectest-F and GiMlf-BAP-rectest-R, and for the presence of the Cas9 with primers Cas9-3500-F and Cas9-pON-R (S4 Table). The encystation of *G. intestinalis* was induced using the Uppsala encystation protocol [65] for the indicated period of time in TYI-S-33 medium (pH 7.8) supplemented with 10% heat-inactivated adult bovine serum (PAA laboratories) and 5 mg/ml of bovine and ovine bile (Sigma-Aldrich–B8381). The encystation efficiency was determined by flow cytometry. The cells were encysted as described above for 48 hours and approximately $1 \times 10^6$ cells were collected by placing them on ice for 10 minutes, centrifuged (1,200 g, 10 min, 4˚C) washed and resuspended in 1 ml of PBS. Antibody against CWP1 protein conjugated with Fluorescein (Waterborne) was added in a dilution of 1:500. Cells were incubated for 1 hour at room temperature in the dark, subsequently washed and resuspended in 1 ml of PBS. The samples were measured using Guava easyCyte 8HT (Luminex) according to previously established flow cytometer settings and gating strategy with a green fluorescence excitation laser (488 nm) and a 530/30 nm detector. The percentage of fluorescent cysts in each culture was calculated. The experiment was carried out in biological triplicate. The mean values were plotted with their standard deviation, and the P-value was calculated by using the two-tailed t-test with equal variance in Excel (Microsoft).

## Growth curves of *Giardia*

To construct growth curves, approximately 100 000 cells/ml of cells in exponential growth phase were inoculated in a total volume of 8 ml of fresh growth medium. Cell concentration was assessed every 24 hours using Guava easyCyte 8HT (Luminex) according to previously established flow cytometer settings and gating strategy, samples were diluted if required. The background signal was measured using fresh medium and subtracted from all samples. Growth curves were plotted as mean values of three biologically independent experiments with standard deviation for each time point using Excel (Microsoft). We have used the CGGC tool [66] to compare the growth curves.

## Cloning and transfection of *Giardia* and *E. coli*

To create the GiMlfendoBAP lineage, the pTGUIDE vector [43] was modified as follows (S2A Fig): the 5' homologous arm was replaced with the sequence of GiMlf-BAP-3UTR by restriction cloning. The sequence GiMlf-BAP-3UTR sequence was first amplified from genomic DNA as two parts, Part 1 (primers GiMlf-part1-MluI-F and GiMlf-part1-BAP-R) encoding the BAP tag at its 3'end and Part 2 (primers GiMlf-part2-BAP-F and GiMlf-part2-AvrII-R) encoding the BAP tag at its 5'end. Equal amounts of both parts were combined into one reaction and 12 cycles of PCR were run without added primers, then we added the GiMlf-part1-MluI-F and GiMlf-part2-AvrII-R primers and ran additional 25 cycles of PCR. The design, hybridization, and cloning of the 746R gRNA into the GiMlf-BAP-pTGUIDE vector was performed as previously described [43]. All used primers are listed in S4 Table. The 3' homologous arm was cut out with ClaI and PacI restriction enzymes, the sticky ends were blunted by incubating the linearized plasmid for 20 min at 72˚C with Q5 polymerase (New England BioLabs) and ligated together. The cell line endogenously tagged with the V5-FAST tag was created in the same manner. Primers V5-FAST-F and V5-FAST-R were used to amplify the tag from a vector carrying the tag as template, GiMlf-NOSTARTrec5-MluIF and GiMlfrec5RV5 were used for

amplification of the 5'homologous arm, and GiMlfrec3-V5-FAST-F and GiMlf-part2-AvrII-R were used for amplification of the 3'homologous arm.

The genes encoding Hsp40 (GL50803_17483), P26s4 (GL50803_113554), Yif1 (GL50803_12945), and ClpB (GL50803_17520) were amplified from genomic DNA with their natural 5'UTRs using forward, and reverse primers listed in S4 Table. The reverse primer of the Yif1 gene contained V5 tag in its sequence and was cloned into the pONDRA plasmid [62] by restriction cloning. The remaining genes were cloned into the resulting V5-pONDRA plasmid in the same way. For the overexpression of Get2 with V5 tag we used already generated plasmid [64]. For the N-terminally V5-tagged Tom40, its 5'UTR was amplified and then linked with amplified V5-Tom40 coding sequence in the pTG plasmid (S4 Table). The Cas9-HA lineage [43] was electroporated with the GiMlf-BAP-746RgRNA-pTGUIDE plasmid using the Gene Pulser (BioRad) as previously described [18]. The expression of the Cas9-HA and GiMlfendoBAP was confirmed by western blot by using rat anti-HA (1:1000, Roche), mouse anti-BAP (1:1000, GenScript)/rat anti-GiMlf antibody (1:1000, [43], respectively. The subcloned lineage GiMlfendoBAP C11 (S2B Fig) was electroporated in the same manner with the Hsp40-V5-pONDRA/ P26s4-V5-pONDRA/ Yif1-V5-pONDRA/ ClpB-V5-pONDRA plasmids. Expression of the gene products of the correct size was confirmed by western blot using the primary rabbit-V5 antibody (1:1000, Abcam).

To be able to express GiMlf-His and its truncated variants, we amplified their sequences from genomic DNA using the primers listed in S4 Table. The sequences were cloned in the pET42b vector by restriction cloning. *E. coli* were transformed with the plasmid using heat shock and positive colonies were selected with the kanamycin antibiotic.

## Confocal fluorescence microscopy, STED, and image processing

The cells were fixed and immunolabeled and image acquisition was performed as described previously [43]. For immunolabeling, commercial primary antibodies mouse/rabbit anti-BAP (1:1000, GenScript), mouse anti-CWP1 (1:500, Waterborne), rabbit anti-V5 (1:1000, Abcam), mouse anti-acetylated tubulin (1:2000, Sigma-Aldrich–T7451), mouse anti-ubiquitin (1:500, Cell Signaling, #3936 –P4D1), self-made primary antibodies rabbit anti-GL50803_9296 (1:2000, [18], rat anti-GiMlf (1:1000, [43] and commercial secondary anti-rabbit/mouse Alexa Fluor 647, anti-rat/mouse Alexa Fluor, and anti-rabbit Alexa Fluor 594 (1:1000, Invitrogen) antibodies were used. To label the PVs, we incubated the cells with dextran-Alexa Fluor 594 (1mg/ml, 10,000 MW, Invitrogen) for 30 min at 37˚C in sPBS as previously described [67], for fixation and subsequent immunolabeling we proceeded with the standard protocol, but permeabilization of the fixed cells was achieved with 0.02% Triton X-100 in PEM buffer. All images were stabilized and deconvolved using the CMLE algorithm with the Huygens deconvolution software. Unless otherwise indicated, the shown images are maximum intensity projections of stack images with adjusted brightness and contrast processed by Fiji ImageJ software [68]. For surface rendering of images with indicated bordering/colocalizing structures, we used the IMARIS software.

For STED microscopy, we used a double concentration of primary antibodies, secondary antibodies anti-mouse STAR 635P (1:100, Abberior) and anti-rat STAR 580 (1:100, Abberior), and Abberior Mount Liquid Antifade medium. The image acquisition was performed as previously described [69].

## Expansion microscopy

Expansion microscopy of *Giardia* was performed as described in [70]. For the immunolabeling, commercial primary antibodies mouse anti-BAP (1:500, GenScript), rabbit anti-V5

(1:500, Abcam), mouse anti-acetylated tubulin (1:1000, Sigma-Aldrich–T7451), self-made primary antibodies rabbit anti-Tom40 (1:1000, [71], rat anti-PDI2 (1:1000, [64], rat anti-GiMlf (1:500, [43] and commercial secondary anti-rabbit/mouse Alexa Fluor 647, anti-rat/mouse Alexa Fluor 488, and anti-rabbit Alexa Fluor 594 (1:500, Invitrogen) antibodies were used. To label the primary amines in cells, we used NHS ester conjugated with Atto 594 (stock solution diluted in DMSO (2mg/ml), final concentration 20 μg/ml diluted in PBS, Merck). We washed the sample labeled with secondary antibody three times for 20 min in MiliQ water, proceeded with the primary amine labeling for 1 hour at room temperature (RT) and washed the sample five times. The expansion factor was experimentally determined to be 3.7. We acquired the images using the Nikon Yokogawa CSU-W1 spinning disc microscope, equipped with PRIME BSI (Teledyne Photometrics) camera and with its own NIS-Elements software. We used the CF Plan Apo VC 60XC WI water objective for acquisition. All images were aligned and deconvolved using the CMLE algorithm with the Huygens deconvolution software and processed as described above.

## Electron microscopy

Cells overexpressing V5-tagged GiMlf and cells with endogenously V5-FAST-tagged GiMlf were fixed in 4% formaldehyde with 0.1% glutaraldehyde in 0.1 M HEPES for 1 h at room temperature (RT). After washing in HEPES buffer, specimens were embedded into 10% gelatin, cryoprotected in 2.3 M sucrose for 48 h at 4˚C, and frozen by plunging into liquid nitrogen. Ultrathin cryosections were cut at -100˚C, and pick-up with 1.15 M sucrose/1% methylcellulose solution (25 cp, Sigma). Sections were incubated for 1h at RT in 1% fish skin gelatin (FSG) in 0,1 M HEPES and labeled with an antibody directed to rabbit monoclonal anti-V5 IgG (1:30, ThermoFisher Sci.) for 30 min at RT. Control sections were incubated without primary antibodies. After washing in 1% FSG, the sections were incubated with protein A conjugated to 10 nm gold nanoparticles (BBI) diluted 1:50 in 0,5% FSG for 1 h at RT. Sections were washed in HEPES, postfixed for 5 min in 1% glutaraldehyde diluted in 0.1 M HEPES, washed in dH2O, and then contrasted/embedded using a mixture of 2% methylcellulose and 3% aq. uranyl acetate solution (9:1). The same procedure was applied to wild type *G. intestinalis* WB c6 to exclude nonspecific binding of the primary anti-V5 antibody.

Samples were inspected with a TEM JEOL 1400 Flash equipped with a CMOS camera EMSIS Xarosa. Tomography data acquisition was done using a TEM JEOL 2100 F equipped with a direct electron detector Gatan K2 Summit and controlled by the SerialEM software package [72]. Two dual-axis electron tomograms were collected: the first within the range of ±65˚ along the x-axis and from -65˚ to 35˚ along the y-axis, the second within the range of ±60˚ along both axes, with a tilt step of 1˚. Tomogram reconstruction was carried out using the IMOD software package [73], which was also utilized for manually masking the area of interest to generate 3D models. 3D model visualizations were performed in IMOD and Amira (Thermo Fisher Scientific) software packages.

## Cell fractionation

*Giardia* trophozoites were grown in two 70 ml cell culture flasks (VWR) to full confluence. The medium was decanted, replaced with chilled PBS buffer and the flasks were placed on ice for 20 minutes to allow cells to detach. The cells were harvested by centrifugation at 1,000 g at 4˚C for 10 min, washed with 15 ml of PBS, then with 8 ml SM buffer (250mM sucrose, 20mM MOPS buffer, pH 7.4) supplemented with cOmplete, EDTA-free protease inhibitors (SM+I, Roche), and resuspended in 800 μl of SM+I buffer. The cells were sonicated on ice four times with 1s pulses and an amplitude of 40 for 1 min (Bioblock Scientific Vibra-Cell 72405). We

kept 100 μl of the lysate as a control. The rest of the lysate was spun at 1000 g for 10 min at 4˚C to remove the unbroken cells. The supernatant was spun twice at 2,680 g for 20 minutes at 4˚C. The pellet (LSP) was resuspended in 100 μl of SM+I buffer. The supernatant was centrifuged at 180,000 g at 4˚C for 30 min. The pellet (HSP) was resuspended in 100 μl of SM+I buffer. The supernatant represented the cytosolic fraction. The protein concentration of the lysate and all collected fractions was measured, and the fractions were diluted to the concentration of 1.6 mg/ml. For the western blot analysis of the fractions, 20 μg of protein were loaded per each fraction. The GiMlf protein was detected with rat anti-GiMlf antibody (1:1000) [43], the mitosomes were detected with rabbit anti-GL50803_9296 (1:2000), the ER was detected with rat anti-PDI2 (1:5000) [64], the cytoplasm was detected with rabbit anti-enolase antibody (1:2000, [64]) and the MT cytoskeletal components were detected by the mouse anti-acetylated tubulin (1:2000, Sigma-Aldrich–T7451).

## Protease protection assay

Cells from one flask were fractionated as described above in SM buffer with 5mM EDTA (SM +E, pH 8), no protease inhibitors were used, and the 2,680 g spinning steps were skipped to obtain a combined fraction of LSP and HSP. The pellet was resuspended in 70 μl of SM+E buffer. Each sample contained 20 μl of pellet suspension, two samples contained trypsin (0.2 μg/ul) and one of them also 0.1% Triton X-100. The samples were diluted with SM+E buffer to a final volume of 50 μl and incubated at 37˚C for 10 minutes. The GiMlf protein was detected with rat anti-GiMlf antibody (1:1000) [43]), the mitosomal matrix protein was detected by the rabbit anti-GL50803_9296 (1:2000), and the mitosomal outer membrane was detected by rabbit anti-Tom40 antibody (1:2000).

## Sodium carbonate extraction

Cells from three flasks were fractioned as described above in SM+I buffer, the 2,680 g spinning steps were skipped to gain a combined fraction of LSP and HSP. The pellet was resuspended in 60 μl of SM+E buffer. We added 200 μl of fresh 100mM $Na_2CO_3$ (pH 11) to 50 μl of the resuspended pellet and mixed the suspension by vortexing. We incubated the mixture on ice for 30 minutes, vortexing it every 5–10 minutes. We added 500 μl of the SM+I buffer and centrifuged the sample at 100,000 g for 30 minutes at 4˚C. The resulting pellet and supernatant were diluted with SM+I buffer to the final volume of 700 μl. The GiMlf protein was detected with rat anti-GiMlf antibody (1:1000) [43]), PDI2 was detected by the rat anti-PDI2 (1:5000), Sec20 was detected by the rabbit anti-Sec20 (1:1000) [43], and VAP by rabbit anti-VAP antibody (1:1000) [30].

## Affinity purification assay

GiMlfendoBAP-CytoBirA and control CytoBirA cells were grown in standard medium supplemented with 50μM biotin for one day prior to harvesting. The cells from eight flasks were fractioned as described above in SM+I buffer, the 2,680 g spinning steps were skipped to gain a combined fraction of LSP and HSP. The pellet was diluted in PBS (pH 7.4) supplied with cOmplete, EDTA-free protease inhibitors (Roche) to the concentration of 1.5 mg/ml. 3 mg of the protein were used per reaction. The protein sample was incubated with 50μM DSP crosslinker (dithiobis(succinimidyl propionate); fresh stock solution diluted in DMSO, Thermo Scientific) for one hour on ice, occasionally mixing by inverting the tube. The experiment proceeded as previously described [12], except the diluted sample was incubated only for one hour at RT while gently rotating with 30 μl of streptavidin-coated magnetic beads (Dynabeads MyOne Streptavidin C1; Invitrogen), and the incubation buffer used for washing contained sodium

deoxycholate (SDC) detergent instead of sodium dodecyl sulphate. One fifth of the sample was used for western blot analysis. Rabbit anti-BAP antibody was used for the detection of GiMlf-fendoBAP in the samples. The rest of the sample was analyzed by MS and the data was processed as previously described [12]. For visualization of results, the VolcaNoseR [74] volcano plot tool was used (P-value = 0.02, fold change $\geq$ 2). The experiment was carried out in triplicate. The localization of each protein was predicted by DeepLoc2 [75]. Prediction of protein/domain homology was performed using the HHpred tool [76], ran against the PDB, SCOPe, and Pfam-A databases.

## Thermal stress induction

*Giardia* trophozoites were grown to full confluence. Cycloheximide (final concentration 100 µg/ml, Sigma-Aldrich 239763) was added to indicated cultures prior heat shock. The cells were placed in a thermal bath set at 43˚C for 20 minutes. The cells were then placed at 37˚C degrees for the indicated time for recovery. The control was placed at 37˚C for the entire experiment. The cells were then processed either for fluorescence microscopy slide preparation as described above or for western blot analysis. For western blot analysis, the cells were put on ice for 10 minutes to detach, pelleted by centrifugation, resuspended in sample buffer (63 mM TRIS pH 6.8, 2.1% SDS, 10.6% glycerol), boiled for 5 minutes at 95˚C and protein concentration was measured with the Bicinchoninic acid kit (Sigma-Aldrich). The samples were diluted to the same concentration and 2-mercaptoethanol was added to 5% final concentration and bromophenol blue was added to 1% final concentration. The sample was again boiled at 95˚C for 5 minutes. The samples were then analyzed using western blot. The GiMlf protein was detected with rat anti-GiMlf antibody (1:1000) [43], rabbit anti-BiP antibody (1:500, commissioned as a custom antibody from Davids Biotechnologie–raised against peptide) and rabbit anti-thioredoxin reductase antibody (1:1000, Davids Biotechnologie–raised against whole recombinant protein purified in our laboratory) was used as a loading control. The densitogram of the peak areas of the detected bands was constructed and analyzed using the Fiji ImageJ software. The resulting values for GiMlf bands were normalized using the values of the loading control.

## Recombinant protein purification and membrane phospholipid affinity assay

To establish the affinity of GiMlf and its truncated variants for membrane phospholipids, we first expressed and isolated the recombinant proteins under native conditions from *E. coli* (NiCo21(DE3), New England Biolabs) transfected with pET42b vectors carrying the His-tagged genes of GiMlf and its variants. The bacteria were grown in large volumes and the protein expression was induced using 0.2mM IPTG for 4 hours at 37˚C while shaking. The bacteria were collected, and the pellet was incubated in N buffer (50mM TRIS, 100 mM NaCl, pH 8) supplied with cOmplete protease inhibitors (N+I, Roche), DNase (1 mg/ml, Sigma-Aldrich) and 5mM $MgCl_2$ and lysozyme (1mg/ml, Sigma-Aldrich) on ice for 30 minutes. The cells were further lysed with a French press and the lysate was spun down for 20 minutes at 100,000 g at 4˚C. All the following steps were carried out at 4˚C. The supernatant was incubated overnight with Ni-NTA agarose resin (Invitrogen) washed with N buffer while gently rotating. The suspension was loaded into a purification column (Invitrogen) and the solution was let to flow through. The column was washed twice with N+I buffer. The proteins were eluted in steps with N buffer with an increasing concentration of imidazole ranging from 5mM to 300mM. For the isolation of the full-length protein and the 54–252 variant, 1M urea was used in all buffers. The eluates were rebuffered to the N buffer using 10kDa 4ml Amicon Ultra

Centrifugal Filter devices (Millipore) and stored at -80˚C with 5% glycerol. The native proteins were then applied on the Membrane Lipid Strip or PIP Strip (Echelon Biosciences) according to the supplied protocol at concentration 2ug/ml. Lipid spots bound by the protein were detected using the rat anti-GiMlf antibody (1:1000).

### Analysis of the GiMlf-KO and GiMlf-V5 overexpression proteomes

The MS proteomic data of the knockout cell line (Cas9-GiMlf-KO H4) and the line overexpressing GiMlf (GiMlf-V5 C5) were analyzed along their corresponding controls (Cas9-pT-GUIDEempty and pTGempty A11). The experiments were performed in biological triplicate and the fold change was calculated by comparing the arithmetic means of the samples and controls. The significance of the results was determined by two-sample test with permutation-based FDR with the value set at 0.05 in the Perseus software [77]. The mass spectrometry proteomics data have been deposited to the ProteomeXchange Consortium via the PRIDE [78] partner repository with the dataset identifier PXD050586. Hypothetical proteins among the significant results were analyzed using the HHpred tool, ran against the PDB, SCOPe and Pfam-A databases. All significant hits were manually assigned functional group as function prediction tools often do not predict a function of hypothetical proteins. The graphic representation of how numerous each group was generated in Microsoft Excel.

## Supporting information

**S1 Fig. (A)** Full phylogenetic tree of GiMlf's homologs from Fig 1 supplied with support values and protein identifiers. **(B)** Multiple sequence alignment of GiMlf and its homologs from different eukaryotic supergroups with indicated secondary structure of GiMlf (α—alpha helix (blue), β—beta strand (yellow), η - 3–10 helix, T–turn). The highlighted positions within the sequence show conservation in at least 70% of the sequences.
(PDF)

**S2 Fig. (A)** Schematic representation of the constructs used for *in situ* tagging of GiMlf with the BAP tag using the CRISPR/Cas9 system. In this case, the *pac* cassette, which provides resistance to puromycin is not part of the repair template. **(B)** Left—PCR amplification of the integrated BAP tag in the genomes of subcloned populations C11, D6, and E7 of *Giardia* modified by CRISPR/Cas9. The Cas9 cell line served as the parental cell line, NC-negative control. Right–PCR amplification of the genomic region containing the *gimlf* gene in modified and control lineages. One of the amplification primers was outside the recombination regions used for the integration of the BAP tag. Only one band was present in all the tested subclones, showing that the BAP tag was integrated into all genomic copies of *gimlf* in the modified lineages. **(C)** Western blot of whole cell lysates of the subcloned endogenously BAP-tagged GiMlf cell line (C11) and a control cell line detecting the expression of Cas9-HA (top panel) with anti-HA antibody, the shift in size and the presence of only a single double band when detected with anti-GiMlf polyclonal antibody (middle panel), and the presence of the BAP tag using anti-BAP antibody.
(PDF)

**S3 Fig.** (A) Wide field image of immunofluorescence analysis of GiMlf (green) and mitosomes (magenta), nuclei stained with DAPI (blue), scale bar 10 μm, red arrowheads highlight the presence of GiMlf at disc margin **(B)** Full images of expansion microscopy images from Fig 2. Scale bars of full images: 10 μm. Scale bars of enlarged sections: 4 μm.
(PDF)

**S4 Fig. Projection of electron tomography without segmentation, white spots correspond to immunolabelling of BAP-tagged GiMlf, scale bar: 250 nm**
(PDF)

**S5 Fig. (A)** Schematic representation of GiMlf-His protein and its truncated forms with the indicated Mlf1IP domain, secondary structure, and positions of truncation of the individual forms. NT–N-terminal domain, CT–C- terminal domain. **(B)** SDS-electrophoresis of purified recombinant proteins stained with Coomassie stain. **(C).** His peptide was used as a negative control for the incubation with the lipid strip, the peptide was detected by an anti-His antibody, TG–triglyceride, DAG–diacylglycerol, PA–phosphatidic acid, PS–phosphatidylserine, PE–phosphatidylethanolamine, PC–phosphatidylcholine, PG–phosphatidylglycerol, CL–cardiolipin, PI–phosphatidylinositol, C–cholesterol, SM–sphingomyelin, 3-SGC– 3-sulfogalactosylceramide, PI(4)P–phosphatidylinositol (4)-phosphate, PI(4,5)P$_2$ –phosphatidylinositol (4,5)-bisphosphate, PI(3,4,5)P$_3$ –phosphatidylinositol (3,4,5)-trisphosphate.
(PDF)

**S6 Fig.** (A) Western blot of the whole cell lysates of the BAP-tagged GiMlf cell line with and without CytoBirA detecting the expression of BirA-HA with anti-HA antibody and the successful biotinylation of BAP-tagged GiMlf using streptavidin. (B) Western blot of the isolation process by biotin affinity purification of BAP-tagged GiMlf and its interaction partners. The ten-fold diluted supernatant of crosslinked proteins was incubated with the streptavidin coated beads. GiMlf-BAP was detected by an anti-BAP antibody. Lys–lysate, P–pellet, S–supernatant, dil–diluted, F–flowthrough, W–wash. (C) Localization of GiMlf and Tom40 in *Giardia* using expansion microscopy. The cells were stained with anti-BAP antibody (green) and anti-Tom40 antibody (magenta). A single layer is shown. Scale bars: 5 μm. (D) The presence of Hsp40 in the axonemes. The cells were stained with an anti-BAP antibody (green) and an anti-V5 antibody (magenta). Nucleic DNA was stained with DAPI (blue), DIC image of corresponding cell is shown in corner of the merged image. Scale bar: 2 μm. (E) Localization of BAP-tagged GiMlf (anti-BAP antibody, green) and its putative interaction partners isolated in the biotin affinity purification assay (anti-V5, magenta) by confocal fluorescence microscopy. DIC image of corresponding cell is shown in corner of the merged image. Scale bars: 2 μm.
(PDF)

**S7 Fig. Localization of GiMlf and V5-tagged P26s4 after heat shock and 4h recovery period using confocal microscopy.** The cells were stained with anti-BAP antibody (cyan), anti-V5 antibody (magenta), and anti-GL50803_9296 antibody (yellow, mitosomal marker). DNA was stained with DAPI (blue). The enlarged images show a single layer of the image stack. Pearson's correlation coefficient ($\rho = 0.5638$) was calculated for the subsection of GiMlf and P26s4 that colocalizes in the proximity of mitosomes, DIC images of the corresponding cells are shown in the corner of the merged images. All scale bars: 2 μm.
(PDF)

**S8 Fig. Fast protein liquid chromatography (FPLC) chromatograms of native GiMlf(ΔCT) recombinant protein (red) and standard protein sample (black) with known molecular weights.** The molecular weight of GiMlf(ΔCT) species was estimated using the logistic model in the CurveExpert software based on the elution volume and the known molecular weight of the standards.
(PDF)

**S9 Fig. (A)** Growth curves of ΔGiMlf and GiMlf-OE cell lines and their respective controls (n = 4). To establish whether the curves statistically differ, the CGGC permutation test (1000

permutations) was used. The growth of ΔGiMlf line was significantly impeded (P-value: 0.024), while there was no significant change in the growth of the GiMlf-OE line (P-value: 0.526) when compared to their respective control cell lines. **(B)** Comparison of the cell morphology of the ΔGiMlf and control cell line using expansion microscopy. Cells were stained with NHS ester, an anti-acetylated tubulin antibody, and an anti-PDI2 antibody. Scale bars: 10 μm. **(C)** Levels of the V5-tagged proteins in the control and ΔGiMlf cells in steady state (control) and after heat shock and 4h recovery. The proteins were detected by an anti-V5 antibody. TrxR was used as loading control. **(D)** Comparison of encysting cells (5 h and 48 h post induction) of ΔGiMlf and control cell lines using confocal fluorescence microscopy. The cells were stained with anti-CWP1 antibody (green). Nucleic DNA was stained with DAPI (blue). DIC images of corresponding cells are shown in corner of the merged images. Scale bars: 2 μm. **(E)** Schematic representation of *Giardia* energy metabolism with indicated downregulated enzymes (red). GK–glucokinase, GPI–glucose-6-phosphate isomerase, PFK–phosphofructokinase, FBA–fructose-bisphosphate aldolase, GA-3-P–glyceraldehyde-3-phosphate, TPI–triosephosphate isomerase, GAPDH–gleceraldehyd-3-phospate dehydrogenase, PGK–phosphoglycerate kinase, PGM—2,3-bisphosphoglycerate-independent phosphoglycerate mutase, ENOL–enolase, PEP–phosphoenolpyruvate, PK–pyruvate kinase, PEPCK–phosphoenolpyruvate carboxykinase (hypothetical protein GL50803_101278, homology inferred from HHpred prediction), PFOR–pyruvate-flavodoxin oxidoreductase, Fdox/red–oxidised/reduced ferredoxin, ADH–alcohol dehydrogenase, AAT–alanine aminotransferase, GluDH–glutamate dehydrogenase, ACS—Acetyl-CoA synthetase, MAL–malic enzyme, MDH–malate dehydrogenase, ATA–aspartate transaminase, GluS–Glutamate synthase, ASNase–L-asparaginase, GPDH–glycerol-3-phosphate dehydrogenase, IPS–inositol-3-phosphate synthase. (PDF)

**S10 Fig.** **(A)** Fluorescence microscopy of the cell line overexpressing V5-tagged GiMlf. Top panel–Presence of large vesicular structures containing GiMlf in the vicinity of mitosomes. Stained with anti-V5 antibody (green) and anti-GL50803_9296 antibody (mitosomal marker, magenta). Nucleic DNA was stained with DAPI (blue). DIC image of corresponding cell is shown in corner of the merged image. Scale bar: 2 μm. Bottom panel–Expansion microscopy of the cell line overexpressing GiMlf-V5. Stained with anti-V5 antibody (green) and anti-PDI2 antibody (ER marker, magenta). Scale bar: 10 μm. **(B)** Transmission electron microscope image of the cell line overexpressing GiMlf-V5 and at low and high magnification, stained with an anti-V5 antibody. The blue arrow points to a peripheral vacuole. Scale bars: 200 nm. (DOCX)

**S1 Table. Table of significantly enriched proteins (P-value < 0.02, Fold change > 2) copurified with GiMlf using affinity purification of biotinylated protein.** Cells expressing only CytoBirA only were used as a control. Homology detection using HHpred was run against the PDB, SCOPe, and Pfam-A databases. (XLSX)

**S2 Table. Table of proteins with significant changes in expression in the GiMlf knock-out cell line compared to the control cell line Cas9-pTGUIDEempty.** In the knock-out line there were 192 significantly upregulated proteins and 307 significantly downregulated proteins, out of which 128 and 205, respectively, were placed within functional groups. The functional group is indicated by color-coded Majority protein IDs (the color of each functional group is indicated in the legend). The significance of the results was determined by a two-sample test with permutation-based FDR with the value set at 0.05. n = 3. (XLSX)

**S3 Table. Table of proteins with significant change in expression in the GiMlf-V5 overexpressing cell line (2.45× upregulation) compared to the control cell line pTGempty.** In the GiMlf-V5 line there were 11 other significantly upregulated proteins and six significantly downregulated proteins, out of which ten and five, respectively, were placed within functional groups. The functional group is indicated by color-coded Majority protein IDs (the colors of each functional group is indicated in the legend). The significance of results was determined by two-sample test with permutation-based FDR with the value set at 0.05. n = 3. (XLSX)

**S4 Table. List of primers used in the study to amplify genomic sequences for the purpose of cloning and to analyze the integration of tags into the genome of *G. intestinalis*.** (XLSX)

## Acknowledgments

The authors acknowledge Imaging Methods Core Facility at BIOCEV, and the Laboratory of Electron Microscopy BC CAS institutions supported by the MEYS CR (LM2023050 Czech-BioImaging, OP VVV CZ.02.1.01/0.0/0.0/18_046/0016045) and for their support & assistance in this work. We would like to thank Ivan Hrdý for the help with FPLC and Lukáš Werner and Jan Štursa for the preparation of lipids.

## Author Contributions

**Conceptualization:** Martina Vinopalová, Pavel Doležal.

**Data curation:** Martina Vinopalová, Lenka Arbonová, Pavel Doležal.

**Formal analysis:** Martina Vinopalová, Lenka Arbonová, Zoltán Füssy, Vít Dohnálek, Marie Vancová.

**Funding acquisition:** Martina Vinopalová, Pavel Doležal.

**Investigation:** Martina Vinopalová, Lenka Arbonová, Zoltán Füssy, Vít Dohnálek, Abdul Samad, Tomáš Bílý, Marie Vancová.

**Methodology:** Lenka Arbonová.

**Supervision:** Pavel Doležal.

**Writing – original draft:** Martina Vinopalová, Pavel Doležal.

**Writing – review & editing:** Martina Vinopalová, Pavel Doležal.

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
