## [Decision Letter · Decision Letter 0]

1 Jul 2024

Dear Dr Dolezal,

Four reviewers have revised your submission. All agree it holds merit and is of value to the community. However, there are a number of comments and observations made by all which warrant a major revision. Please address all major comments for consideration of acceptance.

Thank you,

Carmen Faso

We cannot make any decision about publication until we have seen the revised manuscript and your response to the reviewers' comments. Your revised manuscript is also likely to be sent to reviewers for further evaluation.

Sincerely,

Carmen Faso, Ph.D.

Academic Editor

PLOS Pathogens

Dominique Soldati-Favre

Section Editor

PLOS Pathogens

Michael Malim

Editor-in-Chief

PLOS Pathogens

orcid.org/0000-0002-7699-2064

Dear Dr Dolezal,

Four reviewers have revised your submission. All agree it holds merit and is of value to the community. However, there are a number of comments and observations made by all which warrant a major revision. Please address all major comments for consideration of acceptance.

Thank you,

Carmen Faso

Reviewer's Responses to Questions

**Part I - Summary**

Reviewer #1: The work is very interesting and original, and it contributes to the study in the field of parasitology, particularly regarding the parasite Giardia lamblia. The major strengths are the use of transgenic cells and various techniques to analyze the results. The weaknesses are particularly related to the conclusions drawn from the images showing the localization of the protein, which differ depending on the localization being analyzed. Therefore, I suggest showing photographs of a larger number of cells to trust these results. Additionally, some gels are of low quality, and the entire membrane should be shown. Finally, the study is novel, and if the suggested modifications can be made, it would have a significant impact in the field of giardiasis.

Reviewer #2: Vinopalová et al set out to elucidate the unifying function of Mlf that could explain its phenotypic

diversity. Comparative genomics and Alpha fold demonstrate that Mlf is conserved across the eukaryotic tree. In Giardia MLF was found to localize to basal bodies, the nuclear envelope, the periphery of the ventral disc, and in foci associated with mitosomes and ESVs. Interestingly heat shock upregulated expression of Mlf and the protein was observed to transition from a diffuse distribution to distinct foci. Further over expression of Get2 or Tom40 also caused upregulation of Mlf supporting the idea that Mlf has a role in proteostasis. KO of GiMlf resulted in changes in the levels of proteins regulating proteostasis and a change in the distribution of HSP 40 a conserved Mlf interactor. That GiMlf has a role in proteostasis is convincing. However, the manuscript needs revisions to correct statements about Mlf at MT nucleation zones. Rather it seems Mlf may have a role at the +end of MTs which has to do with growth and stability but not nucleation. Without nucleation we would expect to see a very perturbed MT cytoskeleton in the absence of Mlf, no such phenotype is observed in Mlf KOs.

Reviewer #3: This manuscript investigated the function of the Myeloid leukemia factor 1 in Giardia intestinalis (GiMlf1). The authors used several microscopy techniques (confocal, super resolution, and expansion) to localize giMlf1 more precisely to two areas in the cytosol: 1) microtubule nucleation zones where it interacts with HSP70; 2) in membraneless compartments next to mitosomes where it interacts with regulatory subunit 4 of the 26S proteasome (P26s4). Cellular stress induced by heat shock and by overexpression of organelle-specific membrane protein (Get2 of the ER or Tom40 of the mitosome) results in the relocalization of GiMlf1 from these sites. Interactome analysis provides evidence that GiMlf1 interacts with proteins involved in proteostasis or the regulation of protein quality control in the cell, and they showed colocalization of GiMlf1 with three of these proteins (Hsp40, Tom40, and P26s24). However, they did not see colocalization withYif1, despite it being one of the most highly enriched and most statistically significant proteins identified in the interactome. In general, their microscopy images showing protein colocalization are excellent although the colocalization of GiMlf with the basal bodies and ubiquitin are less convincing. Another lab published a paper (Wu et al., 2021) that showed the involvement of GiMlf1 in autophagy, encystation, and protein clearance in Giardia. Although this study also showed an involvement of GiMlf1 in protein clearance, they showed that a knockout of GiMlf1 likely affects encystation indirectly through disruption of general imbalance in proteostasis that induces a stress response. Furthermore, the authors did not identify proteins related to autophagy in their GiMlf1 interactome obtained from a biotin affinity purification assay with chemical crosslinking. Although the distribution of Mlf1 is found in most eukaryotes, a canonical function has not been determined. This study provides results indicating a role of the Mlf1 homolog in Giardia as a nucleator for protein assemblies to form membraneless compartments in the cytosol to regulate protein folding and degradation; this function may explain diverse Mlf1 phenotypes observed in other eukaryotes including humans.

Reviewer #4: The article titled "Mlf Mediates Proteotoxic Response via Formation of Cellular Foci for Protein Folding and Degradation in Giardia" authored by Vinopalová et al. presents a comprehensive study on the role of Myeloid leukemia factor 1 (Mlf1) homolog (GiMlf) in Giardia intestinalis. Mlf1 is known for its role in hematopoietic differentiation in humans and its involvement in a wide range of other cellular functions, including cell cycle regulation and ciliogenesis. This study aims to elucidate the cellular role of Mlf1 homologs in Giardia intestinalis, focusing on their involvement in protein folding and degradation during proteotoxic stress. The authors specifically present Giardia as a model organism suitable for elucidating the unifying function of Mlf that could make sense of its phenotypic diversity.

The authors show that GiMlf primarily localizes focally to two types of cytosolic structures: microtubule nucleation zones, where it interacts with Hsp40. Ubiquitin-associated, membrane-less compartments containing the 26S proteasome regulatory subunit 4, adjacent to mitosomes. They also provide evidence that GiMlf relocalizes or disperses in response to cellular stress, later forming enlarged foci during recovery. Recombinant GiMlf expressed in E. coli binds to signaling phospholipids on lipid strips, including phosphatidic acid (PA), phosphatidylinositol phosphates (PIPs), and cardiolipin (CL), suggesting a role in membrane recruitment.

They show a possible role of Mlf in proteostasis. Knockout of GiMlf leads to extensive proteomic changes indicative of compromised proteostasis. In addition, ΔGiMlf cells exhibit significant growth impairment, increased cyst production, and altered distribution of interacting proteins (Hsp40 and P26s4).

Taken together, the study underscores the critical role of Mlf proteins in maintaining cellular homeostasis under proteotoxic stress in Giardia by mediating the formation of apparently function-specific foci for protein folding and degradation. The findings provide an advancement in understanding Mlf's cellular functions in Giardia and potentially other eukaryotes. The methodological advance is notable: Utilization of CRISPR/Cas9 in Giardia demonstrated high efficiency in homologous recombination, facilitating the study of gene function without the need for antibiotic selection.

However, there are also some gaps as well as areas that merit much deeper exploration:

The study did not support previous findings on GiMlf's involvement in autophagy and its positive impact on encystation.

There is a lack of observation of membrane-bound GiMlf vesicles and autophagy-related factors in interaction assays.

The precise role of proteasomes in cyst production and viability remains unclear despite their colocalization with GiMlf in ESVs (encystation-specific vesicles).

While the findings are significant for Giardia, the extent to which they apply to other eukaryotes, including humans, remains to be further explored.

The interaction partners identified were not exhaustively validated functionally, leaving room for further investigation into their specific roles and impacts.

The authors performed a convincing phylogenetic analysis across the eukaryotic tree of life showing that the Mlf structure is highly conserved and not surprisingly found in very different roles in different organisms.

The narrative suggests that the goal of the overall investigation was at least in part to identify a unifying role for Mlfs using Giardia as a model (lines 60/61 and 87/88). However, this role has not been identified nor would it from the investigation of its function(s) in an additional, highly diverged eukaryote. Hence, in the light of the distinct possibility of diverse roles in Giardia itself I suggest that the text should be modified to tone down the rationale for this investigation.

**Part II – Major Issues: Key Experiments Required for Acceptance**

Reviewer #1: Due to the fact that the localization of GiMlf changes according to the co-localizations that the authors want to highlight (it is clearly different in Fig 2A and Fig S2D), it would be advisable to include a photograph with a larger number of cells so that the localization of GiMlf can be more conclusively determined.

Line 107: Fig S2D is not sufficient to conclude that the protein associates with PVs, as the resolution of the confocal microscopy is not definitive. Additionally, only one point of co-localization is shown. To assert this, more cells should be shown and perhaps another PV marker should be used, along with expansion microscopy. The DAPI is not visible in the photographs. The same applies to what is described as co-localization with ER. This could be complemented with the use of an anti-BIP antibody, for example.

Supplementary Figure 2: The quality of the Western blot should be improved. Also, the entire membrane should be shown.

Line 108: It is necessary to include more photographs of the different stages of encystation to observe how the localization of the protein changes. Moreover, in Figure 2D, a peripheral localization of GiMlf is observed, which could correspond to PVs or the plasma membrane. To clarify this, the overlap of that image with the DIC should be shown and an anti-PVs antibody should be used. Perhaps during encystation, the protein has a function within these organelles. On the other hand, this localization is not observed in Figure 3E, which shows the different stages of encystation.

Figure 2E should be shown in its original form, and then with the reconstructed cellular structures in color; otherwise, it is impossible to determine the protein's localization.

In Figure 3E, to show the re-localization of the protein, more cells should be shown again, as the pattern of GiMlf distribution, which is generally throughout the cell, is not very clear.

Line 243: Again, this localization in what is called the disk margin is speculative and is not observed in most of the examples presented by the authors. It seems they select the right cell to draw conclusions. To generalize this, more cells should be shown, and co-localization with DIC should be performed.

Lines 258-263 are speculative and should be put in the discussion section.

Figure 6A: The percentage of cysts is lower than generally reported in the literature, both using the two-step protocol and the Uppsala method (30%). The differences may be related to the method used to count the cysts. Generally, cysts cluster together, and when passed through the cytometer, they are discarded as doublets or triplets. I advise counting them using a classical method such as a Neubauer chamber. Additionally, to see if the cysts are viable, particularly in the case of the ΔGiMlf strain, it is advisable to leave them in water for 24 hours and then count them. This is because, under stress conditions, pseudo-cysts often form that do not mature into fully developed cysts.

Reviewer #2: Some specific experiments I would like to see would be 1.) to perform heat shock after treatment with cycloheximide to learn more about whether Mlf foci can re-distribute or if the foci which are often associated with ubiquitin are at the end of their life. 2.) Measure flagella length in control and Mlf KO cells particularly after heat shock. If a phenotype can be found this would support a role for Mlf in regulating MT length/dynamics (more below).

It is said that Mlf associates with MT nucleation sites. This does not appear to be true. Mlf associates with basal bodies but away from where the MT nucleate. Also Mlf associates with the + ends of MTs in the ventral disc. It was incorrectly stated that MTs are nucleated at the disc margin but the Brown reference does not support this. Rather it states “59% of all microtubules in the disc are nucleated at dense bands 39% originate from the disc inner edge, and another 2% appear to have minus ends in the main body of the ventral disc”. Thus no MTs are thought to be nucleated at the disc margin where Mlf localizes. Those are the + or growing ends of MTs. Perhaps Mlf is part of the cap that stabilizes MTs or regulates their length. However not MT defects were observed in the Mlf KOs. Perhaps a MT defect could be uncovered by testing if the Mlf KO is hypersensitive to MT destabilizing drugs.

In several places it is stated that CRISPR KOs or editing can be performed without selectable markers, however selectable markers are used in both PT Guide and the Cas9 vector? I am not sure what the authors are trying to say with their various statements about not needing selectable markers.

Reviewer #3: No new experiments are required. Only clarifications or modifications of the presentation of existing data is requested that will be listed in the next part of this review.

Reviewer #4: Major issues that should be addressed by the authors:

Figure 2A shows an unusual pattern of mitosome morphology with stringy/elongated shapes, instead of the universally observed small spherical structures, like for example in Figure S7A – in addition, there is no data on validation of this antibody. See also below.

Figure 2B/C: expansion microscopy should be accompanied by regular fixation and labeling of cells for comparison which would help to determine whether artefacts are produced. Also, full images of the entire cells including frames for regions of interest should be shown for orientation and to help gauge the reproducibility of the subcellular distribution.

Figure 2E: An overview picture with and without reconstructions should be provided. The question of association of gold particles to specific structures cannot be definitively answered from just one subsection of the cell. Additional imaging data with multiple replicates and transparency about the workflow for segmenting and identifying structures is required to support the interpretation in the text.

Figure 2I: the cloudy background on the delta NT probed lipid membrane on the right is very unfortunate. Also, the interpretation should be that the deletion of the NT disordered region simply partially attenuated the binding strength but not the specificity. Line 891: what was the control?

Most importantly, there is no mention of the CT conferring strict specificity to cardiolipin as shown in Fig S2G! This is in fact what one would expect after deletion of a binding domain as opposed to the mere attenuation elicited after deletion of the NT domain. Cardiolipin is an essential component of mitochondrial membranes which could explain localization of GiMlf to mitosome organelles although assignment of the lipid to mitosomes has yet to be shown.

Lines 135ff : Evidently, a rationale and research question was developed leading to testing the influence of the N-terminal part of Mlf in lipid binding. However, the authors wrap the question (or rather statement) and the answer into one sentence. This also glosses over the fact that deletion of the NT domain yields the same albeit a significantly weakened pattern on the lipid strips in addition to disregarding the key role of the CT for binding to cardiolipin.

Figure 3E: none of the ROIs on the right are in fact enlargements of the merged images – they appear to show a different plane. This begs the question whether the images shown here are indeed “maximum intensity projections” (line 478). This should be fixed. Please note that the same discrepancy is seen in Figure 5A.

Figure 4A: The increase in GiMlf is appreciated, but there is no measure in this figure and 4B of how much overexpression was achieved and the text includes neither this information nor the method with which it was achieved. Both should be incorporated in the main text and 4A and B should be expanded to show baselines of all proteins before and after overexpression.

The quantification of the signals on the Western blot is not valid in my opinion as many of the bands are not in the (relatively narrow) linear range anymore and plateaued. This analysis can only be done with dilution series and measurements in the linear range.

Lines 211/212: First off, a heat shock provoking a heat shock response is kind of a given. .But this also assumes that a mere 6°C difference actually results in massive protein unfolding. Why not make the experiment more specific by using for example DTT to induce a UPR (see e.g. Ravi, Kumar A, Bhattacharyya S, Singh J. 2023 EMBO J.)?

Figure 4C: Documentation of this experiment is insufficient as the bands shown on the Western blot are mostly not in the linear range for densitometric analysis. This experiment should be repeated and expanded to include dilution series and measurements in the linear range for all markers together with the corresponding statistics. Alternatively, a qualitative assessment might be sufficient.

Line 276: What are elongated mitosomes? So far, only spherical organelles have been documented in Giardia. The sentence suggests that there is a population of elongated organelles at steady-state in trophozoites. However, EM studies of the central mitsosome complex has shown consistently that this consists of a grape-like arrangement of individual small spherical organelles embedded withing the basal body complex, whilst small spherical peripheral organelles do not normally aggregate or assume an elongated shape. In this context the bar graph figure in 6C is not quite understandable.

The observed phenotype is striking, however, and indicates a different type of organization of peripheral mitosomes which appear to form linear arrays.

Line 280: Encystation does not appear to be correlated with stress response - at least hallmarks of a stress response due to media changes or protein overload in trophozoites stimulated to encyst are not detected. I believe this has been clarified in Morf et al. 2010. This is actually in line with the ability of trophozoites to persist for weeks in all areas of the intestine using immune-evasion strategies such as antigenic variation. In addition, encystation is not confined to the distal parts of the small intestine where we find a paucity of lipids as the Müller group showed in mice.

Subtle increases in pH such as from the proximal to the distal small intestine and zones of cholesterol depletion are not stressors per se but may trigger signal transduction and elicit a transcriptional response. Hence, there is no evidence for stress as an explanation. Why not just state that this unexpected result requires further research?

Figure 6: Mislocalization is an over-interpretation and implies a loss of function (since Hsp40 is not at its normal site of action anymore). It’s conceivable that Hsp40 relocalizes completely within the range of its intended function to compensate for the absence of GiMlf.

**Part III – Minor Issues: Editorial and Data Presentation Modifications**

Reviewer #1: Line 31: An introductory phrase related to Mlf1 should be added.

Line 47: The citation of Adam is a review. The original paper should be cited.

Line 174: It should say Fig S3D instead of Fig S2D.

In Fig S7A, the letters in figure A are not readable.

Reviewer #2: Line 93: insertion of the (BAP) tag into all four alleles of the gimlf gene without requiring antibiotic selection. Figure S2A indicates that pTguide contains a PAC cassette. So this statement is confusing to me. Do you mean that after editing and selection that integration into the genome allowed maintenance of the strain without antibiotic selection? Or I notice that the schematic shows the Mlf 5’ and 3’ homology regions are only for including the BAP tag, so is it the case that the PAC cassette and gRNA cassette do not get integrated into the genome.

Not a criticism or issue but I am wondering what advantage the BAP tag has over a standard pulldown or TurboID approach. It seems a more complicated approach. It worked so no issue to address, I would just like to understand why one would select this approach?

Figure S2B It would be helpful to add a schematic of where the PCR primers are positioned relative to the tag and chromosomal DNA, perhaps as part of S2A. The results for Mlf in the genome is confusing because this experiment was to edit Mlf with the BAP tag so we expect it to be present. I notice a slight shift so perhaps the point is to show that the tag is present in C11, D6, and E7. Also what was used as the NC?

Figure 2. We were not told how many cells were examined for each localization and if the experiment was repeated multiple times to ensure reproducibility. Is Mlf associated with a subset of mitosomes in every cell or is the association rare. Quantification is important to make this clear. Also if infrequently observed it could just indicate transient interaction. Same issue for 2B, 2C, 2D.

Line 102: The dense band statement is likely wrong. The dense band shown by Hagen should be in the bare area of the ventral disc. Brown 2016 shows there are 6 dense bands and they are very close to the basal bodies. The image is zoomed in so my orientation could be off. Localization to the disc margin and basal bodies is convincing, but more needs to be done to verify dense band localization. Since there are no known markers to be used for co-localization I don’t have any specific recommendation on how to confirm dense band localization. Perhaps the authors can provide additional images that can be compared with the Brown and Hagen results.

Line 104 “margin of the adhesive disc (Fig. 2C,D), where approximately 39% of the microtubules nucleate (Brown et al., 2016)” The Brown 2016 reference is not listed in the bibliography. More importantly, this statement is an incorrect understanding of the literature. The paper indicates that 59% of all microtubules in the disc are nucleated at dense bands 39% originate from the disc inner edge, and another 2% appear to have minus ends in the main body of the ventral disc. So no MTs are reported to be nucleated at the disc margin.

Line 174 “Hsp40 was either present at the axonemes” After looking at Figure 3C it would be more accurate to say at the basal bodies since HSP40 doesn’t mark any other part of the axoneme in WT cells.

Line 178 “the dense band of the adhesive disc (Fig. 3C)” Figure 2C claims to show the dense bands of the disc which I disagreed with above. Figure 3C only shows basal body and disc margin localization? Perhaps this text was meant to be above when discussing Fig 2. However, the authors should drop this statement or provide better images. I think there is good support for Mlf at +ends of MTs but not at -end nucleation sites. In fact, I am unaware of any other MT binding protein associating with both + and – ends.

Line 188 it would be more accurate to say remodeling rather than redesign

Figure 4D is very interesting and I appreciate the quantification. One experiment that could help us understand the relationship between stress and puncta vs ubiquitious distribution would be to treat the cells with cycloheximide before the heat shock. This would block new protein production and allow us to determine if the puncta disperse. Perhaps the puncta are the end of the story and the protein always starts disbursed and then aggregates.

Fig 5A. How many cells were examined and how many independent experiments were performed? Other figures indicate n=50, this is on the lower end for quantification of fixed cells with standard imaging approaches. Our experience has been that values stabilize after about 200 cells. I suspect if another 50 cells are quantified the presented values would be different.

Line 252 here a comma is used instead of a period and in other places a comma is used to indicate thousands. Edit this instance for consistency.

Line 258 “indicates that no GiMlf-positive, membrane-bounded vesicular structures of appropriate size could be observed by electron microscopy after immunogold labeling” I don’t think the authors can be confident about this statement. Consider that there are only a handful of foci per cell and data is presented for phospholipid binding which would indicate association with membranes. How many sections were analyzed? Was the EM performed on heat shock recovered cells that were shown to have a large increase in Mlf foci?

Figure 6B. The caudal flagella appear longer in control cells after heat shock recovery. This should be quantified by measuring flagella length. If true it would help point towards a function at flagella tips (+ ends of MTs).

Line 355 I don’t believe basal body localization should be equated with the MT nucleation zone. It seems that Mlf is not at the business end. Also there is discussion of Mlf’s role in cilliogenesis, but GiMlf was never demonstrated to be in the axonemes, we only saw that HSP40 is at the + ends of MTs after heat shock in the Mlf mutant. Perhaps Mlf has an indirect role by regulating HSP40 which does have a role in cilia motility (Zhu 2019 MBOC).

Line 378 indicates that KOs are made without selectable markers but selectable markers are used. This statement makes no sense to me.

Line 425 punctuation

Antibodies listed in the methods: The specific antibody name should be provided to avoid ambiguity- For example Cell Signaling has multiple anti-ubiquitin antibodies and Sigma has multiple anti-tubulin antibodies.

Reviewer #3: 1. Fig. 1 and Fig. S1: it is strange that GiMlf1 groups with the Euglenozoa in the phylogenetic trees. The authors should include a comment about this. Also, do Trichomonads and Entamoeba have MLF1 homologs?

2. Fig. 2: the colocalization of GiMlf1with the basal bodies is not very clear even in the expansion microscopy images. Is there a reason why the authors did not attempt to colocalize the GiMlf with centrin since a centrin antibody (Millipore 04-1624) that works in Giardia is available?

3. Lines 106-107: "a minor fraction of the protein was associated with other membrane-bound organelles including the nuclear envelope (Fig. 2C)." I am not sure if the association of GiMlf1 around the nuclear envelope should be considered “minor”. This adverb should be removed.

4. Lines 216-217: "increase in GiMlf protein levels was more gradual, reaching a peak at 4 hpi., and could only be detected after 30 minutes of the recovery period". This sentence needs to be corrected as Fig. 4C shows a band for GiMlf even at 0 h.

5. Lines 216-264: "It is plausible that these foci are in fact built by the oligomerized GiMlf subunits as high molecular weight species corresponding to GiMlf octamers, as observed by size exclusion chromatography of purified recombinant GiMlf(ΔCT) (Fig. S5)". How did they get an octamer from the size exclusion that shows peaks at 36 kDa and 163 kDa for GiMlf(ΔCT). The form at 163 kDa is closer to a tetramer.

6. Lines 284-285: "The amounts of the V5-tagged proteins were comparable to those in control cells, but their cellular distribution was significantly altered in the ΔGiMlf cell line." Where is the western blot that show the levels of the V5-tagged proteins in ΔGiMlf is the same as those in the control cells?

7. Line 413. The catalog number of the bovine bile used for encystation should be given as different types of bovine bile are available from Sigma and they have differences in their ability to induce Giardia encystation.

8. Line 413: the efficiency of encystation was determined by flow cytometry. These % efficiencies should be given.

9. Line 582: "Giardia trophozoites were grown in 10ml plastic tubes with flat bottom (Thermo Scientific) to full confluence". Why were “flat bottom” tubes mentioned? Is this important for the thermal stress experiment? If so, they should say why and give catalog numbers for these tubes. A better way of phrasing this would be to say "were grown in flat-bottomed 10 mL plastic tubes".

10. Growth curves were plotted with cell counts from using a using Guava easyCyte 8HT (Luminex). Were the cells stained with a viability dye or other dyes for counting?

11. More information about the source of some of the antibodies is needed. Where or how were the antibodies for PDI2, enolase, GiMlF1, Bip, and thioredoxin reductase obtained? Were they made in the authors’ lab, obtained from another lab, or commissioned as a custom antibody from a company? What were the antigens used: – peptide or recombinant protein?

12. Line 891: CT – control – what was the control? There is no sample labeled as CT in Fig. 2I.

13. Fig. 4C: why did the authors choose to use thioredoxin reductase as a loading control in the western blot? Did they try to use common housekeeping proteins such as actin as a loading control?

14. Line 1055: 10xx diluted. Typo needs to be fixed to read "ten-fold"

15. Supplementary Table 2 Table of proteins with significant changes in expression in the GiMlf knock-out cell line compared to the control cell line Cas9-pTGUIDEempty. What is the “NaN” entry in the “fold-change” column? Are these proteins identified in the KO cell lines that are NOT present in the control cell lines? Therefore, these are proteins unique to the KO cell lines.

Reviewer #4: Minor Issues:

Line 128: The title should be revised to fit with the text: e.g. Recombinant Mlf binds structural and signaling membrane lipids in vitro.

Line 129f: Given the data presented up to this point, the claim is rather surprising. The authors should explain.

Figure S6F is mislabeled

Lines 133/34: The sentence is unclear and should be revised.

Lines 366f: That's an appealing idea, but not supported experimentally in this work. It has been shown that stress response and encystation are not coupled at least in the sense that the former does not induce encystation and the latter does not elicit a stress response. The informative experiment would be to test whether the rate of spontaneous encystation is increased in KO cells. This would require complementing ectopically with a dd-tagged Mlf gene and stabilization with Shield/Rapamycin to reverse the effect.

Lines 358ff: The authors state “Considering our data alongside these independent findings, it seems plausible that the unifying function of Mlf proteins is to mediate the formation of concentrated function …”. They set out to identify a unifying function in a poorly characterized model system. This conclusion is premature and excludes many possible other scenarios.

PLOS authors have the option to publish the peer review history of their article (what does this mean?). If published, this will include your full peer review and any attached files.

Reviewer #1: No

Reviewer #2: No

Reviewer #3: No

Reviewer #4: No
---

## [Editor Report · Decision Letter 1]

26 Sep 2024

Dear Dr. Doležal,

We are pleased to inform you that your manuscript 'Mlf Mediates Proteotoxic Response via Formation of Cellular Foci for Protein Folding and Degradation in Giardia' has been provisionally accepted for publication in PLOS Pathogens.

Best regards,

Carmen Faso, Ph.D.

Academic Editor

PLOS Pathogens

Dominique Soldati-Favre

Section Editor

PLOS Pathogens

Michael Malim

Editor-in-Chief

PLOS Pathogens

orcid.org/0000-0002-7699-2064
---

## [Editor Report · Acceptance letter]

11 Oct 2024

Dear Dr. Doležal,

We are delighted to inform you that your manuscript, "Mlf Mediates Proteotoxic Response via Formation of Cellular Foci for Protein Folding and Degradation in Giardia," has been formally accepted for publication in PLOS Pathogens.

Best regards,

Michael Malim

Editor-in-Chief

PLOS Pathogens

orcid.org/0000-0002-7699-2064